# Modeling the topographic influence on aboveground biomass using a coupled model of hillslope hydrology and ecosystem dynamics

Yilin Fang[1], Ruby Leung[1], Charlie Koven[2], Gautam Bisht[1], Matteo Detto[3], Yanyan Cheng[4], Nate McDowell[1,5], Helene Muller-Landau[6], S. Joseph Wright[6], and Jeff Chambers[2]

[1]Pacific Northwest National Laboratory, Richland, WA, United States
[2]Lawrence Berkeley National Laboratory, Berkeley, CA, United States
[3]Department of Ecology and Evolutionary Biology, Princeton University, Princeton, NJ, United States
[4]Department of Industrial Systems Engineering and Management, National University of Singapore, Singapore
[5]School of Biological Sciences, Washington State University, Pullman, WA, United States
[6]Smithsonian Tropical Research Institute, Balboa, Panama

*Correspondence to*: Yilin Fang (yilin.fang@pnnl.gov)

**Abstract.** Topographic heterogeneity and lateral subsurface flow at the hillslope scale of ≤ 1km may have outsized impacts on tropical forest through their impacts on water available to plants under water stressed conditions. However, vegetation dynamics and finer-scale hydrologic processes are not concurrently represented in Earth system models. In this study, we integrate the Energy Exascale Earth System Model (E3SM) Land Model (ELM) that includes the Functionally-Assembled Terrestrial Ecosystem Simulator (FATES), with a three-dimensional hydrology model (ParFlow) to explicitly resolve hillslope topography and subsurface flow and perform numerical experiments to understand how hillslope scale hydrologic processes modulate vegetation along water availability gradients at Barro Colorado Island (BCI), Panama. Our simulations show that groundwater table depth (WTD) can play a large role in governing aboveground biomass (AGB) when drought-induced tree mortality is triggered by hydraulic failure. Analyzing the simulations using random forest (RF) models, we find that the domain-wide simulated AGB and WTD can be well predicted by static topographic attributes including surface elevation, slope and convexity, and adding soil moisture or ground water table depth as predictors further improves the RF models. Different model representations of mortality due to hydraulic failure can change the dominant topographic driver for the simulated AGB. Contrary to the simulations, the observed AGB in the well-drained 50-ha forest census plot within BCI cannot be well predicted by the RF models using topographic attributes and observed soil moisture as predictors, suggesting other factors such as nutrient status may have larger influence on the observed AGB. The new coupled model may be useful for understanding the diverse impact of local heterogeneity by isolating the water availability and nutrient availability from the other external and internal factors in ecosystem modeling.

## 1 Introduction

The aboveground biomass (AGB) within forests is a large storage pool for carbon, so reliably quantifying the spatial distribution of AGB is important for understanding the role of forests in the carbon cycle and in climate change mitigation [*Garcia et al.*, 2017; *Hernandez-Stefanoni et al.*, 2020; *Houghton et al.*, 2009]. The spatial distribution of AGB is commonly acquired from remote sensing or extensive field collection of plot data [*Benitez et al.*, 2016; *Condit et al.*, 2019; *Goita et al.*, 2019; *Goncalves et al.*, 2017; *Hernandez-Stefanoni et al.*, 2020; *Hernandez-Stefanoni et al.*, 2018; *Zaki and Abd Latif*, 2017; *Zald et al.*, 2016]. However, it is challenging to understand the dynamic structure and biomass of forests and how they may respond to climate change, especially for tropical forests with high tree diversity [*Clark et al.*, 1999; *Feroz et al.*, 2014; *Wiegand et al.*, 2017].

One factor that could play an important role in organizing the spatial distributions of tropical tree species is habitat variability, such as topographic conditions, soil biotic and abiotic characteristics, and soil water levels [*Costa et al.*, 2005; *Echiverri and Macdonald*, 2019; *Grasel et al.*, 2020; *Kinap et al.*, 2021; *Mascaro et al.*, 2011; *Miron et al.*, 2021; *Oliveira et al.*, 2019; *Schietti et al.*, 2014; *Steidinger*, 2015; *Zuleta et al.*, 2020]. Analyses of the spatial patterns of tropical species have shown that topographic attributes, such as slope and curvature, are a strong driver in controlling AGB variation in tropical forests [*Detto et al.*, 2013; *Mascaro et al.*, 2011; *Silveira et al.*, 2019]. However, the mechanisms responsible for the association between topography and forest structure are not well understood. For example, soil moisture varies strongly with topography, and several studies have demonstrated how drought-associated mortality, species composition, structure and functions are all dependent on soil moisture gradients and water table depth [*Schietti et al.*, 2014; *Terra et al.*, 2018].

Previous ecosystem dynamics modeling studies have included foci on non-spatial species distribution, statistical species-area relationship, and spatially explicit trees [*Fisher et al.*, 2018; *Moorcroft et al.*, 2001; *Sato et al.*, 2007; *Schumacher et al.*, 2004; *Wiegand et al.*, 2017; and references therein]. However, they largely ignored hillslope hydrological processes, which fundamentally modulate water, energy, and biogeochemical fluxes at local scales [*Fan et al.*, 2019]. A quantitative assessment of the influence of hillslope water availability on ecosystem dynamics has not been undertaken, partly due to limited availability of observational data and

limited capabilities of models to represent processes at relevant scales. Our aim for this study is to develop a new modeling capability that incorporates the forest response to variation in

hillslope soil moisture content and water table dynamics in an Earth system modeling framework. While ecosystem dynamics models have been coupled with land surface models, the latter generally ignore hillslope hydrologic processes or represent them crudely using subgrid parameterizations [*Clark et al.*, 2015]. More detailed hydrologic models that represent hillslope hydrology and subsurface processes have been coupled to land surface models, but ecosystem

dynamics models have not been included in those land surface models [e.g., *Kollet and Maxwell*, 2006]. Models such as Regional Hydro-Ecologic Simulation System (RHEESyS) [*Tague and Band*, 2004] and Terrestrial Regional Ecosystem Exchange Simulator (TREES) [*Mackay et al.*, 2015] can represent vegetation dynamics with hillslope hydrology, but they have not been incorporated in Earth system models for modeling the coupled Earth system processes. In a

comparison of a land surface model with a three-dimensional hydrology model in the Asu catchment of the Amazon basin, Fang et al. [2017] found significant influence of topography on groundwater table and runoff. Without subsurface lateral flow, the land surface model cannot reproduce the seasonal dynamics of the groundwater table simulated by the three-dimensional hydrology model. Hawthorne and Miniat [2018] suggested that through redistribution of soil

moisture, topography may mitigate drought effects on vegetation along a hillslope gradient. It is recommended by Swetnam et al. [2017] that the non-linear effects of lateral redistribution of water in complex terrain should be taken into account to improve the prediction accuracy of tree mortality. These motivate the need to model hillslope hydrologic processes and ecosystem dynamics in a single Earth system modeling framework, as the seasonal dynamics of water

available to plants could have significant effect on plant growth and survival during drought.

To develop a new modeling capability to study the role of hillslope water availability on ecosystem dynamics, we couple the land component of the Energy Exascale Earth System Model (E3SM) [*Golaz et al.*, 2019; *Leung et al.*, 2020] in a configuration that includes a vegetation demographic model called the Functionally-Assembled Terrestrial Ecosystem Simulator

(FATES) [*Huang et al.*, 2020; *Koven et al.*, 2020; *Negron-Juarez et al.*, 2020; *Powell et al.*, 2018], with a three-dimensional hydrology model (ParFlow) [*Ashby and Falgout*, 1996; *Jones and Woodward*, 2001; *Kollet and Maxwell*, 2006; *Maxwell*, 2013]. The goal is to provide a tool in the Earth system modeling to isolate the plant water availability from the other controlling

factors associated with topography for AGB variability. The coupled model developed in this study is used to evaluate the role of hillslope water availability to ecosystem functioning at Barro Colorado Island (BCI), Panama, where observations of both vegetation and hydrology are available. BCI exhibits higher aboveground biomass on slopes and wet swamp [Chave et al., 2003]. Furthermore, higher mortality rate of canopy trees at a plateau in BCI during 1983 was attributed to water stress by low precipitation and high temperature [Condit et al., 1995]. To our knowledge, no coupled modeling of ecosystem dynamics and hillslope hydrology has been conducted at the site.

Hydraulic failure is the inability of a plant to move water from roots to leaves. It is one of the physiological mechanisms for tree mortality [*McDowell et al.*, 2011]. Observed and projected increases in drought frequency, intensity, and duration increased the risk of hydraulic failure and vulnerability of trees [Allen et al., 2015]. We hypothesize that hydraulic failure induced mortality has a significant impact on AGB variability along the hillslope hydraulic gradient. In this study, we conduct numerical experiments using the newly developed coupled model to investigate how model structure (i.e., model with or without lateral flow captured by ParFlow), plant functional composition (represented by different functional traits in FATES), as well as alternative methods representing hydraulic failure induced mortality can influence ecosystem dynamics at BCI. We briefly summarize each model, followed by a description of the approach used to couple the models. We then describe a set of numerical experiments and compare the model simulations with field observations. To evaluate the influence of topography on AGB through its impact on hydrologic processes, we analyze and compare the simulations across the model domain to determine the sensitivity of the simulated AGB to model structure, plant functional composition, soil property, and representations of hydraulic failure. Lastly, we develop random forest (RF) models using various topographic attributes and the simulated and observed soil water states as predictors to predict the simulated and observed AGB. The purpose of the RF models is to reveal whether there are similar nonlinear relationships between topography, soil water states, and AGB in the coupled simulations and in the observations. This analysis may reveal model limitations in capturing certain nonlinear relationships found in the observations and inform future efforts to improve modeling of coupled hydrology-vegetation processes.

## 2 Methods

### 2.1 Model descriptions

To achieve the goals of this study, we used the land model of E3SM called ELM, the integrated hydrology model called ParFlow capable of simulating surface and subsurface flow at hillslope scale, and the FATES vegetation demographic model to develop a coupled model of vegetation-hydrology interactions at hillslope scale. The model components and the coupling approach are described below.

#### 2.1.1 The Energy Exascale Earth System Model (E3SM) Land Model (ELM)

The Energy Exascale Earth System Model (E3SM) is an Earth system model containing modules for land, ocean, sea ice, and river [*Caldwell et al.*, 2019; *Leung et al.*, 2020]. The land model in E3SM, referred to as ELM, started as a branch of the Community Land Model version 4.5 (CLM4.5) [*Oleson et al.*, 2013]. The one-dimensional model simulates changes in canopy water, surface water, snow water, soil water, soil ice, and water in the unconfined aquifer through parameterization of interception, throughfall, canopy drip, snow accumulation and melt, water transfer between snow layers, infiltration, evaporation, surface runoff, sub-surface drainage, vertical redistribution within the soil column, and groundwater discharge and recharge [*Oleson et al.*, 2013]. The default soil hydrology model in ELM solves the one-dimensional Richards' equation in unevenly spaced vertical soil layers. The solution of the Richards' equation is driven by precipitation, infiltration, subsurface runoff, evaporation, and canopy transpiration through root extraction, and interactions with groundwater. Water flux input to the ground surface (the top grid cell surface), is the liquid water reaching the ground, which is then partitioned between surface runoff, surface water storage, and infiltration into the soil. Runoff generation in ELM can be parameterized using either the TOPMODEL-based [*Beven and Kirkby*, 1979] runoff model (SIMTOP) [*Niu et al.*, 2005] or the runoff parameterization of the Variable Infiltration Capacity (VIC) model [*Liang et al.*, 1994]. Soil hydraulic properties are determined according to sand and clay contents based on the work by Clapp and Hornberger [1978] and Cosby et al. [1984], and organic properties of the soil [*Lawrence and Slater*, 2008].

#### 2.1.2 ParFlow

ParFlow solves the following Richards' equation in variably saturated soils in three dimensions [*Kollet and Maxwell*, 2006; *Kuffour et al.*, 2020]:

$$S_s S_w(h) \frac{\partial h}{\partial t} + \phi \frac{\partial S_w(h)}{\partial t} = -\nabla[k_s k_r(h)\nabla(h+z)] + q_s \tag{1}$$

where $t$ is time (s), $S_s$ is the specific storage (m$^{-1}$), $S_w$ is the relative saturation [-],$\phi$ is the effective porosity of the media, $h$ is pressure head (m), $k_s$ is the saturated hydraulic conductivity tensor (m h$^{-1}$), $k_r(h)$ is the relative permeability [-], $z$ is the elevation (m), and $q_s$ is the source term (h$^{-1}$). The saturation-pressure and relative permeability-saturation functions can be represented by either the van Genuchten [1980] or the Brooks and Corey relationship [*Brooks*

*and Corey*, 1966]. The following simplified Brooks and Corey relationship is used in this study:

$$\frac{\theta - \theta_r}{\phi - \theta_r} = \left(\frac{p_a}{p}\right)^\lambda \tag{2}$$

$$k_r = \left(\frac{\theta - \theta_r}{\phi - \theta_r}\right)^n \tag{3}$$

where $\theta$ is water content, $\theta = \phi\, s(p)$, $\theta_r$ is the residual water content, $\lambda$ is the pore size distribution index, $p_a$ is the bubbling capillary pressure, and $n$ is the pore disconnectedness index,

which equals $3+2/\lambda$.

ParFlow has an integrated overland flow simulation capability, where a free-surface overland flow boundary condition is applied at the land surface and overland flow is solved with the kinematic wave equation [*Kollet and Maxwell*, 2006]. At the top boundary between the surface and subsurface systems, pressure continuity between the two systems is assigned. Only

when the top cell of the subsurface domain is ponded is the kinetic wave equation activated [*Maxwell et al.*, 2016]. One of the model options we use in this study is the terrain following grid (TFG) [*Maxwell*, 2013] capability to define the gridded domain to conform to topography, which is useful for coupled surface-subsurface flow problems. When discretized numerically into grids in three dimensions, Eq. 1 equates the time rate of change of water mass within a grid with

the mass fluxes of water across the surfaces of each grid as well as water source/sink. This results in a matrix equation including every grid, both horizontally and vertically. The water table is the surface where the water pressure head is equal to the atmospheric pressure. The surface was calculated by the hydraulic head of the water saturated (i.e., the soil moisture equals

the porosity) grid near the ground surface. The Richards' equation is solved numerically using cell-centered finite difference in space and an implicit backward Euler scheme in time [*Kollet and Maxwell*, 2006]. It is designed for high performance applications and is solved using a parallel, globalized Newton method and a multigrid-preconditioned linear solver [*Ashby and Falgout*, 1996; *Jones and Woodward*, 2001].

### 2.1.3 The Functionally Assembled Terrestrial Ecosystem Simulator (FATES)

The Functionally Assembled Terrestrial Ecosystem Simulator (FATES) is a cohort model of vegetation competition and co-existence that was originally separated from the ecosystem demography model in the community land model (CLM(ED)) [*Fisher et al.*, 2015], which was based on the ecosystem demography concept in Moorcroft et al. [2001]. The tiling structure in FATES represents the disturbance history of the ecosystem via dynamically tracking areas with similar disturbance histories, which are referred to as 'patches', replacing the plant functional type (PFT) structure in the organization hierarchy in CLM. The patch has no spatial location association. In doing so, FATES uses a given "Host Land Model". Currently supported host land models are the Community Land Model of the Community Terrestrial Systems Model (CLM-CTSM) and E3SM Land Model (ELM). Boundary conditions are clearly identified between FATES and the host land models where FATES functions are invoked [*Koven et al.*, 2020].

Figure 1 shows the information that is passed between FATES and ELM at each ELM model step (half-hourly) for biophysics and at the end of each day for vegetation dynamics. At each ELM time step, ELM provides FATES with environment conditions (e.g., soil moisture, atmospheric forcing etc.), and FATES calculates surface processes and provides ELM terms (e.g., canopy conductance, albedo, leaf area index, root water extraction to meet transpiration demand, etc.) to calculate canopy level fluxes. Daily cohort-level carbon increment or net primary productivity (NPP) is used to allocate carbon to plant organs and alter the cohort structures. At the daily time step, daily carbon increment calculated in FATES is sequentially allocated per cohort [Koven et al., 2020]. The amount is subtracted from the cohort's storage pool if the carbon increment is negative. If the carbon increment is positive, the cohort first replenishes the carbon storage pool and tissue turnover is then compensated. The cohort will allocate the remaining increment to any organ pools (leaf, stem, coarse root, fine root, and seed) that are below their allometric targets. The cohort will grow its stem diameter, allocating to each pool proportionally

to that pool's derivative with respect to stem diameter using the remaining carbon increment (if

any). Patch structures can also be altered by disturbance processes from fires, small-scale tree mortality, and anthropogenic disturbance.

FATES uses allometric relationships with stem diameter ($D$) to determine tree height ($h$) and crown area ($C$). There are five model options for tree height in FATES. In this study, we used a power function described in Obrien et al. (1995):

$$h = 10^{(log10(D_*)\cdot a + b)} \tag{4}$$

$$D_* = \min\left(D, D_{max}\right) \tag{5}$$

and a Michaelis–Menten form in Martinez Cano et al. (2019):

$$h = \frac{cD_*^d}{k + D_*^d} \tag{6}$$

The allometry function for crown area is

$$C = \begin{cases} fD^g & D < D_{max} \\ fD_{max}^g & D \geq D_{max} \end{cases} \tag{7}$$

where a, b, c, d, k, f, and g are allometric parameters, Dmax is diameter of plant where max height occurs.

Target biomass (the projected quantity along the tangent of the allometric curves from

where they started) of leaf, structure, stem, fine root, seed, and storage are also calculated using allometry functions in FATES (Koven et al., 2020). Target biomass of fine root and storage are assumed to be linearly proportional to the target leaf biomass, and the target structure biomass is linearly proportional to the target sapwood biomass.

A power law allometric model is used for the target leaf biomass (L):

$$L = mD_*^g \tag{8}$$

where $m$ and $g$ are allometric parameters, and $g$ is the same as in Eq. 7.

FATES has three allometry function options to calculate target stem aboveground biomass ($C_{agb}$), we used the functional form in Saldarriaga et al. (1998):

$$C_{agb} = f_{agb} p_1 h^{p_2} D^{p_3} \rho^{p_4} \tag{9}$$

and a functional form in Chave et al. (2014):

$$C_{agb} = \frac{1}{c2b} p_1 (\rho D^2 h)^{p_2} \tag{10}$$

where $f_{agb}$ is the fraction of stem above ground, $p_1$, $p_2$, $p_3$, and $p_4$ are allometry parameters, $c2b$ is carbon to biomass ratio, $\rho$ is the plant wood density.

Once tissue turnover and storage carbon demands are met, FATES uses a constant fraction of net primary production for seed production. Total aboveground biomass (AGB) reported in the study is the sum of leaf biomass, aboveground stem biomass and seed biomass.

Total plant mortality per cohort is simulated as the sum of the six additive terms including mortality due to carbon starvation and hydraulic failure [*McDowell et al.*, 2011], fire, size, age, and background mortality that is unaccounted by any of the other mortality rates. Among these mortality mechanisms within the model, we are particularly interested in the mortality induced by hydraulic failure as we expect different vegetation response to plant water availability along the hillslope.

The default hydraulic failure model in FATES uses a proxy for hydraulic failure induced mortality. For each day, mortality with a rate $M_{hf,coh}$ is triggered (or a set fraction of trees are killed) if the plant wilting factor is beyond a threshold (default is $10^{-6}$ (unitless)) using the following equation:

$$M_{hf,coh} = \begin{cases} m_{ft} & \text{for } \beta < 10^{-6} \\ 0.0 & \text{for } \beta \geq 10^{-6} \end{cases} \tag{11}$$

where $m_{ft}$ is a constant specific to a plant functional type, $\beta$ is the water stress factor that depends on soil water matric potential as follows [*Oleson et al.*, 2013]:

$$\beta = \sum_i \frac{\psi_C - \psi_{S,i}}{\psi_C - \psi_O} r_i \tag{12}$$

where $\psi_{S,i}$ is the soil water matric potential in soil layer $i$ (m), $r_i$ is the root fraction in soil layer $i$, $\psi_C$ is the soil water potential (m) when stomata are fully closed, and $\psi_O$ is the soil water potential

(m) when stomata are fully open. $\beta = 1$ when vegetation is unstressed, and $\beta = 0$ when the plant wilting point is reached. The threshold value of $10^{-6}$ represents a state where the average soil moisture potential is within $10^{-6}$ of the wilting point. As a default option in FATES, when $\beta$ is below this threshold, a set fraction of the tress with rate $M_{hf,coh}$ (yr$^{-1}$) is killed as a proxy for hydraulic failure induced mortality.

Alternatively, a mechanistic hydraulic failure model is based on the plant hydraulics model in FATES, i.e., FATES-hydro, where hydraulic failure mortality begins when plant fractional loss of conductivity (*ftc*) reaches a threshold (*ftc,t*, default is 0.5):

$$M_{hf,coh} = \begin{cases} \frac{ftc-ftc,t}{1-ftc,t} m_{ft} & \text{for } ftc \geq ftc,t \\ 0.0 & \text{for } ftc < ftct \end{cases} \tag{13}$$

where $m_{ft}$ is the maximum mortality rate (yr$^{-1}$). FATES-hydro solves the water transport through different organs in the plants, from roots to leaves. It considers the plant internal water storage, which can buffer the imbalance of root water uptake and transpiration demand. Details of FATES-hydro can be found in Christofferson et al. [2016] and Fang et al. [2021].

We also tested another hydraulic failure model assuming the drought mortality rate as a linear function of soil water potential using, for example, the slope derived in Kupers et al. [2019a] based on the observations of first year mortality rate of naturally regenerating seedlings to soil water potential for one species from the study site:

$$M_{hf,coh} = b \psi_S \tag{14}$$

where b is a constant (b = 0.49 yr$^{-1}$ MPa$^{-1}$), $\psi_S$ is soil water potential (MPa).

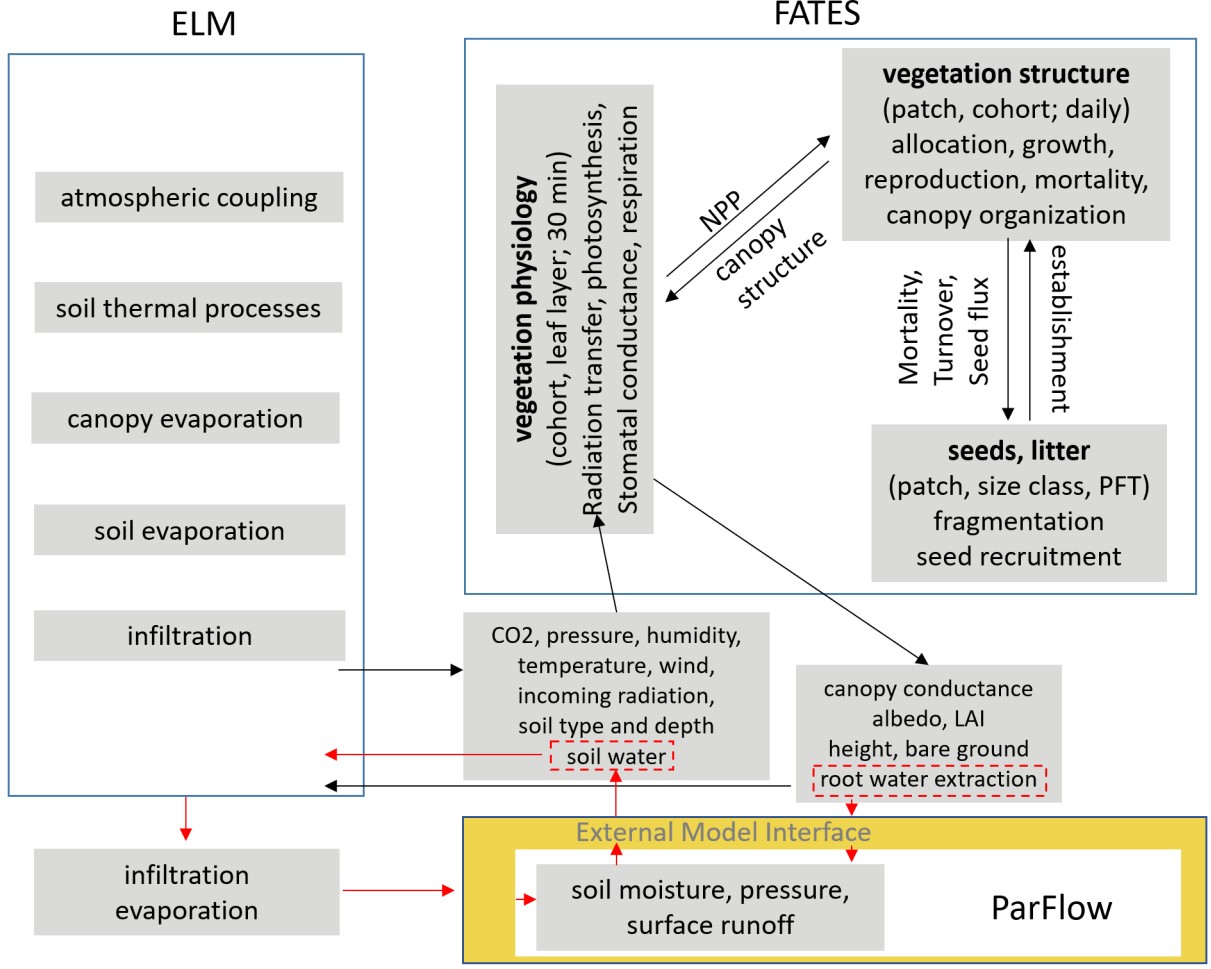

**Figure 1**. Schematics of ELM, ParFlow and FATES and the approach to couple the three models. Hydrology in ELM is replaced by ParFlow. Arrows show the passing of variables between models. Black arrows indicate the exchange of variables within FATES and between ELM and FATES. Red arrows highlight the exchange of variables between ELM and ParFlow. Interactions between FATES and ParFlow are mediated through ParFlow's influence on soil water and FATES' influence on root water extraction, shown in the red dashed boxes.

## 2.2 Model coupling approach

ParFlow was previously coupled to version 3.5 of CLM or CLM3.5 to simulate physical processes related to the energy and mass balance at the land surface [*Maxwell and Miller*, 2005]. Many changes have been made relative to CLM3.5 ever since then in terms of processes and modularized code structure. CLM3.5 was not designed to host FATES because of its code structure. Instead of modifying CLM3.5, the ELM and ParFlow coupling approach in this study

combines the approaches used to couple the land model and the subsurface model adopted by Maxwell and Miller [2005], Kollet and Maxwell [2006], and Bisht et al. [2017]. Coupling is achieved by: (1) replacing the one-dimensional models for flow in unsaturated and groundwater zones in ELM by ParFlow to simulate unsaturated–saturated flow within the three-dimensional subsurface domain, (2) replacing the runoff scheme in ELM with the integrated overland flow module in ParFlow, and (3) providing ELM with the soil moisture simulated by ParFlow (Fig. 1) at each time step.

ParFlow is incorporated in ELM in a distributed manner as a module through an external model interface (EMI). Only vegetated surfaces are allowed in this coupling such that each tile in ELM coincides with the upper face of the uppermost cell (ground surface of the subsurface computational domain) in ParFlow using a terrain following grid. In other words, each vertical column of the ParFlow grids corresponds to a soil column in ELM. The decomposition approach for ELM and ParFlow are round-robin decomposition and domain decomposition, respectively. Therefore, mapping of gridded data from one model onto the grids of the other is required through sparse matrix vector multiplication based on preprocessed sparse weight matrices between the two models [*Bisht et al.*, 2017]. For simplification, the size of soil columns of the two models are the same, i.e., the elements in the sparse weight matrices are 1.0. The new namelist "use_parflow_emi" in the land model is required to run the coupled model. As shown in Fig. 1, for each time step, ParFlow receives infiltration, evaporation, and root water extraction from ELM and provides its calculated soil moisture to ELM through the model coupling interface. Note that FATES does not have direct interface with ParFlow. The effect of ParFlow on FATES is through the soil moisture it passes to ELM, and the effect of FATES on ParFlow is through the root water extraction it passes to ELM, as indicated by the dashed red boxes in Fig. 1.

## 2.3 Site description and observation data

Our model experiments are conducted at Barro Colorado Island (BCI) (9°10'N, 79°51'W), Panama, which is one of the world's best-studied tropical forests [*Leigh*, 1999] because of the century-long presence of a research station and ongoing scientific investigation focused mainly on terrestrial forest ecology and related fields [*Wright*, 2020]. BCI is administered by the Smithsonian Tropical Research Institute (STRI). After canal construction

and the formation of Gatún Lake in the Chagres River in 1914, BCI became isolated from the surrounding mainland [*Zimmermann et al.*, 2013]. It rises out of the waters of the man-made Lake Gatún (normal water level of 26 m above sea level) and has an area of 1560 ha which is

covered by forest that has remained relatively undisturbed for at least 100 years (https://biogeodb.stri.si.edu/physical_monitoring/research/barrocolorado). The two main geological formations at BCI are the Bohio from the early Oligocene and the younger Caimito formation from the late Oligocene, both are sedimentary rocks consisting of volcanic and marine facies [*Grimm et al.*, 2008 and references therein]. The clay-rich Cambisols and Ferralsols

dominate the soils at BCI and the mean soil textures largely belong to silty loam, silty clay, clay, and clay loam textural classes [*Grimm et al.*, 2008]. Measured saturated hydraulic conductivity at the site varies from 0.016 to 13.2 mm/h [*Kinner and Stallard*, 2004].

The site has long-term meteorological and hydrological data. Meteorological data from 2003-2016 is available from a meteorological tower near the Lutz catchment at BCI

[*Faybishenko et al.,* 2018]. The wet season at BCI is roughly from May to December and the dry season is from late December to April. Annual mean precipitation during the simulation period is 2382.7 mm, while mean precipitation in the dry season is 219 mm. Observed evapotranspiration (ET), gross primary production (GPP), sensible heat flux (SH), and latent heat flux (LH) at the site was obtained from an eddy-covariance system installed in July 2012 on the AVA tower

(~1.25 km from the Lutz catchment) located 41m above the ground on the top plateau. Locations of the Lutz tower and the AVA tower are shown in Fig. 2. Three Time Domain Reflectometers (TDR, CS616, Campbell Scientific) were installed vertically in the vicinity of the AVA tower in July 2012. The apparent dielectric permittivity of soil measured by TDR probes is related to the soil water content using an *ad hoc* calibration curve [*Kelleners et al.,* 2005] using seven *in*

*situ* gravimetric soil water content samples (0-15 cm) collected near the probes during different soil moisture regimes (30 campaigns).  The 50-ha permanent plot on BCI (1000 m × 500 m) was established in 1981. Censuses have been carried out in 1981-1983 and every five years from 1985 to 2015. In each census, all woody stems at least 1 cm diameter-at-breast-height were identified, measured, and mapped. Over 350,000 individual trees have been tallied over 35 years

[*Condit et al.*, 2012; *Condit et al.*, 2019; *Condit et al.*, 2017; *Hubbell and Foster*, 1983]. The aboveground biomass along with a 5 m topography survey of the BCI 50-ha plot by Harms et al. [2001] can be found in the 2019 version the BCI forest census plot database [*Condit et al.*, 2019].

Maps of soil water potential and soil water content for several dry season stages during 2015 and 2016 in the 50-ha plot were generated by Kupers et al. [2019b] based on measurements of a total

of 1299 samples at a total of 363 sites that covered all soil types and habitats in the plot area. Most samples were taken at the 15 cm depth.

## 2.4   Numerical experiments

Figure 2 shows the ParFlow simulation domain and the surface elevation at the site, as well as the 50-ha forest dynamics plot (consisting of quadrats of 5 m by 5 m). The ParFlow domain is

selected to minimize the boundary effect on the flow within the 50-ha plot, by providing a buffer between the edge of the ParFlow domain boundary and the 50-ha plot boundary. The elevation in the study domain ranged from ~28 to 186 m above sea level, with a moderately gentle topography [Lobo and Dalling, 2013]. The model is driven by the same atmospheric forcing (i.e., precipitation, air temperature, relative humidity, wind speed, and surface pressure) for 2003-

2016 measured at a meteorological tower near the Lutz catchment at BCI [*Faybishenko et al., 2018*] in all grids due to the lack of spatial forcing. Comparison of the precipitation at the tower and a clearing near the Lutz catchment shows good agreement supporting the use of the same atmospheric forcing for all grids of the model.

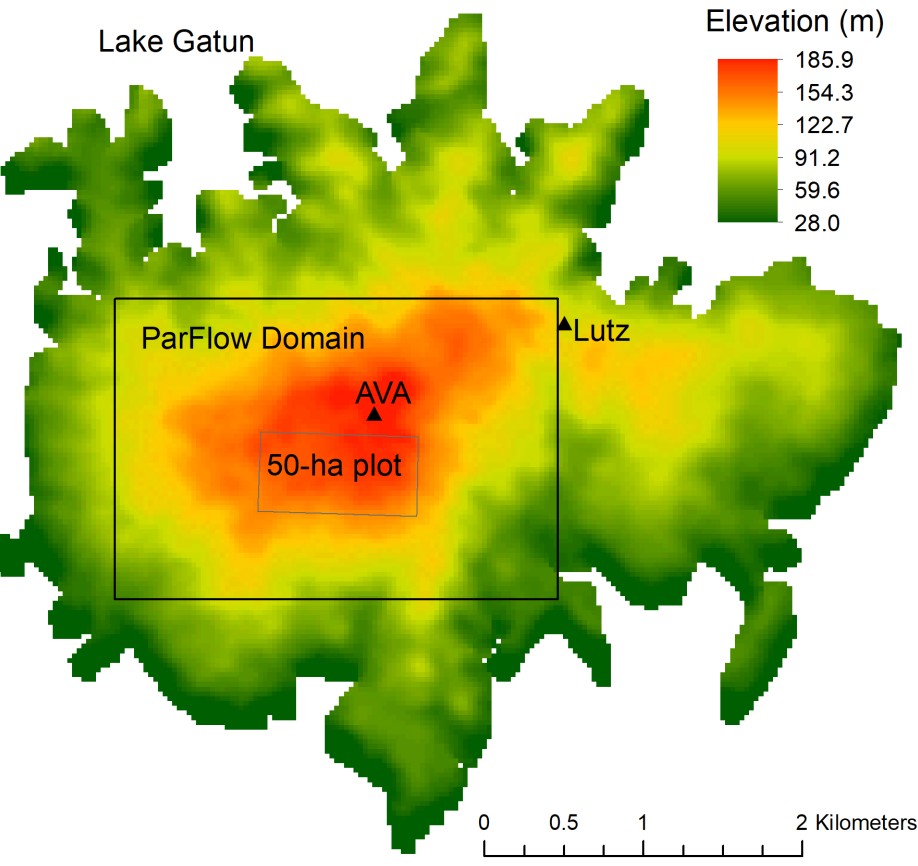

**Figure 2**. Simulation domain and elevation. The black rectangle inside is the ParFlow simulation domain, and the smaller grey rectangle indicates the 50-ha census plot on the highland. Locations of the AVA eddy covariance tower and the Lutz meteorological tower are shown by the small triangles.

Seven model experiments (Table 1) are conducted to evaluate model sensitivity to model structure, plant traits, soil property, and the hydraulic failure representations. Specifically, two of the experiments are run using ELM-FATES without ParFlow to evaluate sensitivity to model structure (Cases 1 and 2). The other five simulations are run using ELM-ParFlow-FATES with different combinations of plant traits, soil property, and representations of tree mortality rates due to hydraulic failure. The reasons for these selected simulations are that 1) plant traits directly affect vegetation structure and water use, 2) soil property affects WTD, thus plant water availability; 3) elevated mortality rate for canopy trees at BCI was observed during the severe dry season of 1983 [Condit et al., 1995], which can be triggered by hydraulic failure.  Soil saturated hydraulic conductivity and saturation function parameters for ParFlow are calculated

from ELM based on soil texture and organic matter content. Another set of soil water retention

parameters was derived from soil water potential data in Kupers et al. [2019b]. As there are no

site-wide groundwater table measurements, for simplicity no-flux boundary conditions are

applied at the bottom boundary and the lateral boundaries of the ParFlow simulation domain

assuming they have minimal impact on the results at the 50-ha plot in the center of the domain,

at least 0.5 kilometer away from the lateral boundaries as the 50-ha is in the high elevation zone

of the domain. The number of grids in x, y, and z direction are 31, 21, and 15, respectively. The

grid resolution for ParFlow in the x and y directions is 90 m and varies from 7 mm (near the

ground surface) to 35 m (near the bedrock) in the z direction. The 30 m resolution digital

elevation model (DEM) of the Republic of Panama, generated by NASA SRTM program is

aggregated and smoothed using cubic convolution resampling technique to 90 m resolution to

calculate the slopes for the ParFlow simulations.

In FATES, plant functional types (PFTs) are represented by a vector of plant traits. All of

the numerical experiments are initialized with equal low number density of seedling (0.2

individuals/m$^2$) of broadleaf evergreen tropical PFT and are spun-up for 100 years using ELM-

FATES, without ParFlow. Model comparisons are based on the results for another 100 years

after the spin-up for Cases 1 to 4, and additional 16 years for Cases 3, 6 and 7 for hydraulic

failure model comparisons starting from the 200-year result of Case 3. The 16-year simulation

period was chosen such that the meteorological forcing aligns with the years of observations.

Another cycle of forcing was run for Case 4 using soil property derived from Kupers' to get

results of Case 5. If not noted, results reported in this study are based on the corresponding

simulation years after the spin-up. Two PFTs representing early successional and late

successional species are simulated at the same time in competition with each other using two

input files of plant traits selected from previous ensemble simulations that best matched

observations for tropical forests [Chen et al. 2022, Huang et al. 2020]. Further parameter tuning

is out of the scope of this work. Those ensemble simulations were used to examine the sensitivity

of tropical forest dynamics to hydrological and physiological parameters. The two input files we

use contain trait parameters for both early and later successional species, and they are referred to

as F1 and F2, respectively. F1 and F2 differ in vegetation biomass allometric models and

parameters, as well as the fraction of woody biomass that is aboveground and mortality rate from

carbon starvation. The allometric models for tree height and target stem aboveground biomass in

F1 are defined in Eqs. 4 and 9, respectively, and those in F2 are defined in Eqs. 6 and 10, respectively. F2 has a smaller maximum carbon starvation mortality rate ($S_{m,ft}$) and larger aboveground woody biomass fraction compared to F1. The complete parameters for F1 and F2 are included in the Supplement. In FATES, the actual carbon starvation mortality ($M_{cs,coh}$) is calculated as a function of the non-structural carbon storage ($C_{store,coh}$) and the PFT-specific 'target' leaf carbon ($C_{leaf,target}$) as

$$M_{cs,coh} = \max\left(0.0, S_{m,ft}\left(0.5 - \frac{C_{store,coh}}{C_{leaf,target}}\right)\right) \tag{15}$$

Three drought mortality models M1, M2, and M3 corresponding to Eqs. (5), (6), and (7), respectively, are evaluated. FATES-hydro is turned off for models M1 and M3. Details of each case are described in Table 1.

## 2.5  Random forest models

Topography attributes have previously been found to influence soil water, groundwater depth and vegetation structures [*Condon and Maxwell*, 2015; *Detto et al.*, 2013; *Holyman et al.*, 2018; *Lan et al.*, 2011; *Mascaro et al.*, 2011; *Pachepsky et al.*, 2001; *Sener et al.*, 2005; *Tai et al.*, 2020; *Zinko et al.*, 2005]. As the relationships between AGB, hydrologic processes, and topographic attributes are likely complex and nonlinear, we develop RF regression models to evaluate how well static topographic attributes and hydrologic states may be used to predict the AGB in observations and model simulations. Such analysis can be used to determine how well the nonlinear relationships in observations may be captured by the coupled model and whether the RF models may be used as a more computationally efficient approach to represent the nonlinear relationships simulated by the complex models. To evaluate which topographic attributes (land surface elevation (DEM), slope, and Laplacian convexity) have more controls on plant water availability and aboveground biomass, we develop RF models using monthly output at each grid from our coupled model in year 2015 (a year when observations were also available) for Cases 3, 5 and 6, based on a supervised machine learning module from the Scikit-learn machine learning library in Python [Pedregosa et al. 2011]. The analyses are performed both domain-wide and for the 50-ha plot (Fig. 2). Variables that are simulated based on modeling of physical processes are also used as predictors to evaluate RF model accuracy. Similar analysis is performed for the observations in the 50-ha plot using the AGB, and soil moisture estimated

based on measurements across the plot, and the 5 m DEM grid from the 2019 version of the BCI

database. The spatial soil water content across the plot in Kupers et al. [2019b] are linearly interpolated, and AGB are aggregated at each of the 5 m DEM grid location for the analysis.

The slope and convexity are computed from the first and second order derivatives of the smoothed DEM (z) that's aggregated for the 90 m resolution as follows [*Detto et al.*, 2013]:

$$slope = \arctan\sqrt{f_x^2 + f_y^2}\,; f_x = \frac{\partial z}{\partial x}; \ f_y = \frac{\partial z}{\partial y} \tag{16}$$

$$convexity = f_{xx} + f_{yy}; f_{xx} = \frac{\partial^2 z}{\partial x^2}; \ f_y = \frac{\partial^2 z}{\partial y^2} \tag{17}$$

Positive convexity values are in the areas of depressions and valleys, and negative values in peaks or ridges.

For each RF model based on the simulated results and observation in year 2015, 75 percent of the data are allocated to the training set and 25 percent to the test set. Hyperparameters of the

RF models are selected using the scikit-learn's function "RandomizedSearchCV" [Pedregosa et al. 2011]. Permutation importance, which measures the increase in model error or how much the model depends on a feature when the relationship between the feature and the target is broken, is reported for each RF model. To calculate the permutation importance, a reference score (prediction error) for a trained regression model *m* is first calculated. Each feature *j* (a column)

in the training or testing dataset is randomly shuffled to generate a corrupted dataset and the score of the model *m* on the corrupted dataset is calculated. The shuffling and corrupted dataset score computation are repeated multiple times. The importance of feature *j* is computed as the difference between the reference score and the arithmetic mean of the scores of the model *m* on the corrupted datasets. This is documented in https://scikit-learn.org/stable/about.html#citing-

scikit-learn.

The performances of the RF models are quantified using the mean absolute percentage error (MAPE) and percent of variance explained ($VAR_{ex}$):

$$MAPE = \frac{1}{n}\sum_{i=1}^{n}\left|\frac{y_{i,pred}-y_i}{y_i}\right| \times 100\% \tag{18}$$

$$VAR_{ex} = \left(1 - \frac{\sum_{i=1}^{n}(y_{i,pred}-y_i)^2}{\sum_{i=1}^{n}(y_i-\bar{y_i})^2}\right) \times 100\% \tag{19}$$

**Table 1**. Definition of model experiments with ELM, PF, F, and M denoting E3SM land model, ParFlow, different parameters for plant traits, and different mortality models, respectively. K in experiment name of Case 5 indicates soil property derived from Kupers et al. [2019b] is used. Extra 16 years of simulation were conducted for four experiments. K in experiment name of Case 5 indicates soil property derived from Kupers et al. [2019b] is used.

| Cases | Model Experiments | Plant Traits | Soil Property | ParFlow | Drought Mortality Model | Extra simulation years for model comparison |
|---|---|---|---|---|---|---|
| 1 | ELM-F1-M1 | F1 | S1 | No | Eq. (11) | 0 |
| 2 | ELM-F2-M1 | F2 | S1 | No | Eq. (11) | 0 |
| 3 | ELM-PF-F1-M1 | F1 | S1 | Yes | Eq. (11) | 16 from Case 3 |
| 4 | ELM-PF-F2-M1 | F2 | S1 | Yes | Eq. (11) | 0 |
| 5 | ELM-PF-F2-M1,K | F2 | S2 | Yes | Eq. (11) | 16 from Case 4 |
| 6 | ELM-PF-F1-M2 | F1 | S1 | Yes | Eq. (13) | 16 from Case 3 |
| 7 | ELM-PF-F1-M3 | F1 | S1 | Yes | Eq. (14) | 16 form Case 3 |

## 3   Results

### 3.1   Model sensitivity to lateral flow representation

This section focuses on model sensitivity analysis as no spatial observations are available for comparison with the model simulations. Averages for the year 2015 for selected variables are plotted in Fig. 3 for ELM-F1-M1 and ELM-PF-F1-M1, to assess the spatial impact of lateral flow on these variables. Results from ELM-PF-F1-M1 exhibit the largest spatial variability in terms of ground water table depth (WTD), vegetation biomass, and heat fluxes, showing large gradients between plateau and valley. Lacking representations of lateral flow (case ELM-F1-M1) results in less spatial variability in those variables of interest (Fig. 3a,c,e,g). ELM-F1-M1 simulates shallower water table depth below the ground surface and lower Bowen ratio (the ratio of sensible to latent heat fluxes) at the plateau compared to the lowland (Fig. 3a,g). For ELM-PF-F1-M1, wetter soil at lowland favors higher latent heat flux and smaller sensible heat flux,

resulting in smaller Bowen ratio compared to the plateau area (Fig. 3h). In ELM-PF-F1-M1, the simulated ground water table elevation generally follows the topography. There is a sharp transition in AGB and GPP associated with the large hydraulic gradients or sharp transition of ground water table depth above and below ~5 m at lowlands, with wetter area having larger AGB and GPP. Note this is based on model comparisons. Spatial observations at those locations are needed to validate the model but such observations are not currently available. But away from the transition zone, AGB and GPP are relatively insensitive to WTD in these model configurations (Fig. 3d,f).

The simulated AGB is 3 times smaller than the observed AGB (15.5 kg C/m$^2$ assuming a conversion factor of 0.5 from dry weight to carbon equivalents) in 2015 using the plant traits F1 and two times smaller using plant traits F2 (Fig. 4e). With a main interest in the spatial variability of AGB and without model calibration to reduce differences between simulations and observations, we compare the observed AGB and the simulated AGB using normalized values (scaling to unit norm). Standard deviations of the normalized AGB at the 50-ha plot for ELM-F1-M1 and ELM-PF-F1-M1 are 0.008 and 0.014, respectively. They are smaller than 0.072 calculated from the observed AGB aggregated to the simulation grids. From Fig. 3a,b, variability of the normalized WTD from the simulations are 0.011 and 0.16 for ELM-F1-M1 and ELM-PF-F1-M1, respectively at the 50-ha plot, higher than the variability of simulated AGB. The spatial correlation between modeled and observed biomass is not significant inside the 50-ha plot for all of the simulations because of homogeneity of the meteorological forcing, soil properties, and gentle topography. This suggests that WTD is not the dominant controlling factor for AGB at the plot based on correlation analysis and model assumptions.

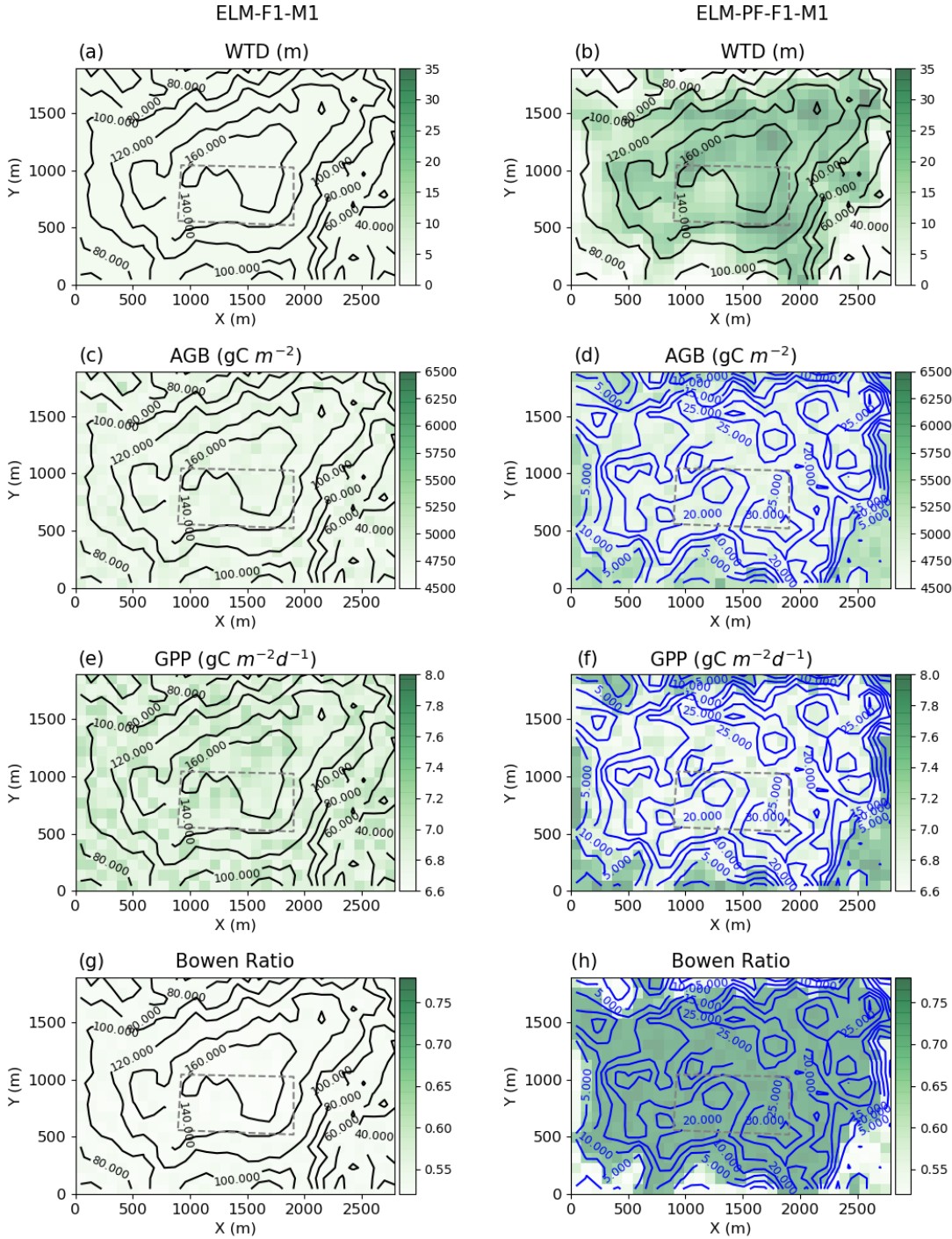

**Figure 3.** Sensitivity of model predictions to lateral flow dynamics and water table depth. Water table depth (WTD) (a,b), aboveground biomass (AGB) (c,d), GPP (e,f), and Bowen ratio (g,h) for ELM-F1-M1 (no lateral flow), and ELM-PF-F1-M1. The blue contour lines in d, f, and h are WTD, and the black contour lines in the rest are ground surface elevation (m). The 50-ha plot is located in the region within the dashed line.

## 3.2   Influence of model configurations

Model experiments with the plant traits F1 result in the survival of only early succession trees. Here we evaluate and compare model simulations with F1 and F2 in different model configurations to evaluate the impacts of the latter. Across the various simulations shown in Table 1, simulation ELM-PF-F1-M1 shows the maximum spatial standard deviations of variables of interest (shaded area in Fig. 4). From that simulation, the spatial standard deviation (STD) of monthly gross primary productivity (GPP) is 1.42 ($g/m^2/d$) (Fig. 4a), latent heat flux (LH) is 19.5 ($W/m^2$) (Fig 4 b), sensible heat flux (SH) is 17.2 ($W/m^2$) (Fig. 4c), volumetric water content (VWC) in the top 15 cm of soil is 0.084 ($m^3/m^3$) (Fig. 4d), AGB is 0.28 (kg $C/m^2$) (Fig. 4e), and WTD is 13.7 (m) (Fig. 4f). For each month, standard deviations are calculated based on the spatial variability within the simulation domain and the monthly maximum standard deviations are determined by comparing the standard deviations across the model simulations. However, even the largest variability of AGB is only 5.5% of the average AGB while the VWC variability can be as large as 21% of the average VWC. WTD is deeper and has a large seasonal variability when lateral flow is represented in simulations with Parflow. The large differences of spatial average of GPP, LH, SH among simulations in the wet season are caused by plant functional traits, while the differences of VWC and WTD, and land surface fluxes in the dry season are caused by lateral flow representation (Fig. 4). In general, the simulated GPP and LH center around the observations, while the simulated SH and VWC are biased high and low, respectively, compared to the observations. As sensible heat flux is negatively related to soil moisture, it can be improved by a better parametrization of soil moisture dynamics, for example, by using different soil properties in the model as will be shown later. The model was not able to capture the temporal dynamics of GPP, it's not clear what's the cause. Model parameters and

measurement uncertainty can both contribute to the biases. This is a model limitation that needs to be addressed in the future.

Using plant trait F2, ELM-PF-F2-M1 generates a forest of coexisting early succession and late successional PFTs. The spatial standard deviations (STDs) of the aforementioned-variables of interest for simulation ELM-PF-F2-M1 are slightly smaller than for ELM-PF-F1-M1. The difference in STD between ELM-PF-F2-M1 and ELM-PF-F1-M1 is larger for VWC, LH, and SH compared to GPP, AGB and WTD. With this plant traits F2, AGB increases by

47.5% and GPP decreases by 19% on average (Fig. 4). As the soil moisture (VWC) simulated using ELM-PF-F1-M1 and ELM-PF-F2-M1 are close (Fig. 4d), GPP is mainly affected by growth allometry while AGB is the result of both growth and mortality. Using plant traits F2 results in larger growth rates and significantly lower mortality rates (Fig. S1), thus increases of AGB for F2 compared to F1. However, simulation with F2 results in much lower exposed leaf

area index, thus lower GPP compared to that with F1. Based on the model results, species competition also cannot explain the observed variance of AGB at the 50-ha plot without accounting for the spatial heterogeneity of soil properties, nutrient availability, plant traits, etc. in the model. For example, wood density can contribute to the observed variability as it is a parameter used to define the allometry function (Eq. 10). AGB can be further increased by

parameter tuning, but we don't expect it to significantly change the AGB variability.

Model structure (ELM vs ELM-PF) and soil property have larger effect on soil water than on energy, carbon fluxes and AGB and vice versa for plant traits (Figs. 4 and 5). Using soil water retention curve from Kupers et al. [2019b] improved wet season soil moisture, dry season sensible heat flux, and GPP, as well as some of the observed peak GPP in wet season. It also

significantly changed WTD compared to the simulation with the original soil property (Fig. 5f). The soil moisture in the dry season was overestimated, possibly due to the no-flux boundary conditions that created overall wetter soil in the domain at areas adjacent to the boundaries.

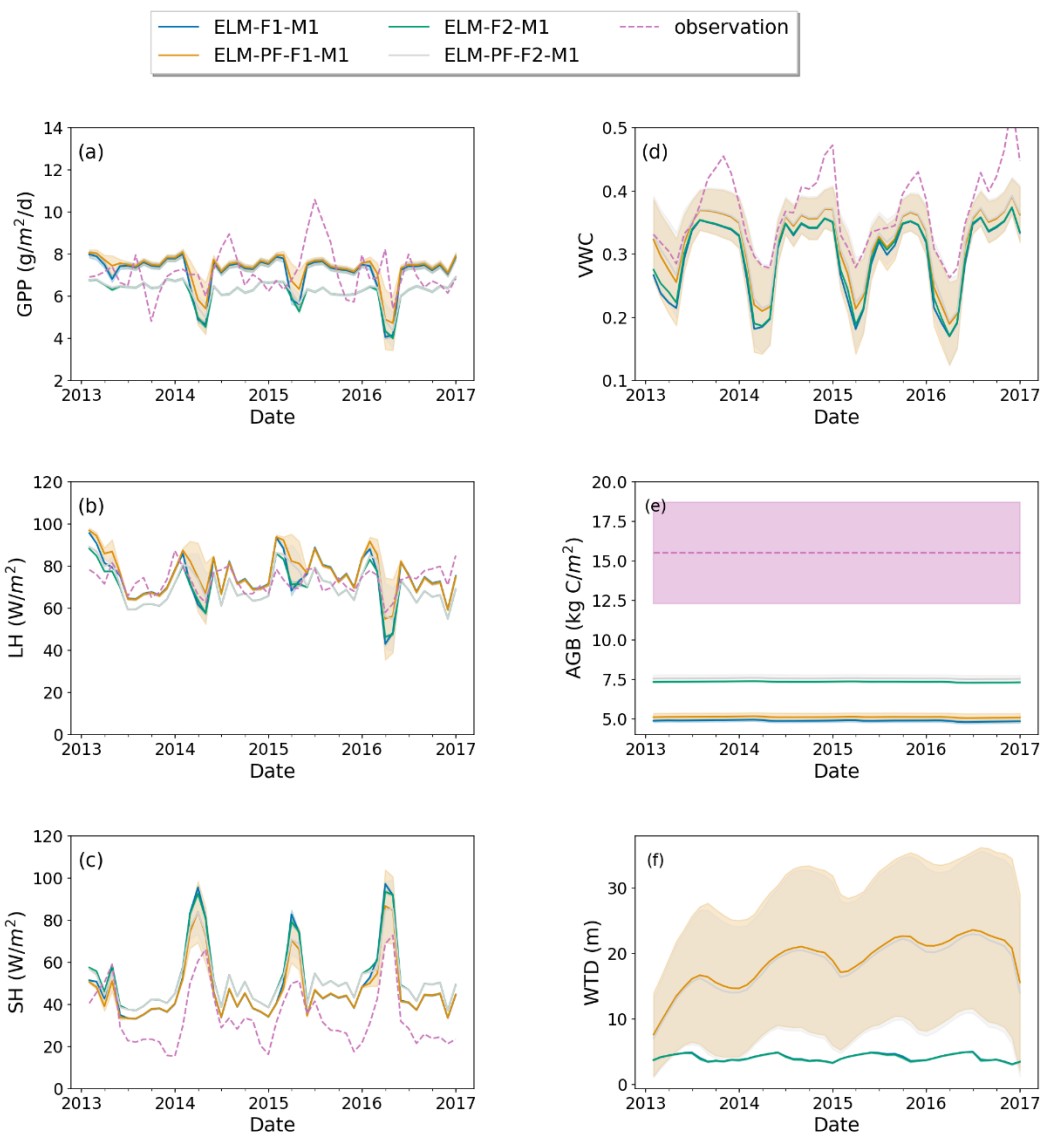

**Figure 4**. Simulated GPP (a), latent heat flux (LH) (b), sensible heat flux (SH) (c), top 15 cm volumetric water content (VWC) (d), aboveground biomass (AGB) (e), and groundwater table depth (WTD) (f) for simulations ELM-F1-M1, ELM-F2-M1, ELM-PF-F1-M1, and ELM-PF-F2-M1. Dashed line is the observation if available. Solid line is spatial average and shaded area is the standard deviation over the simulation domain.

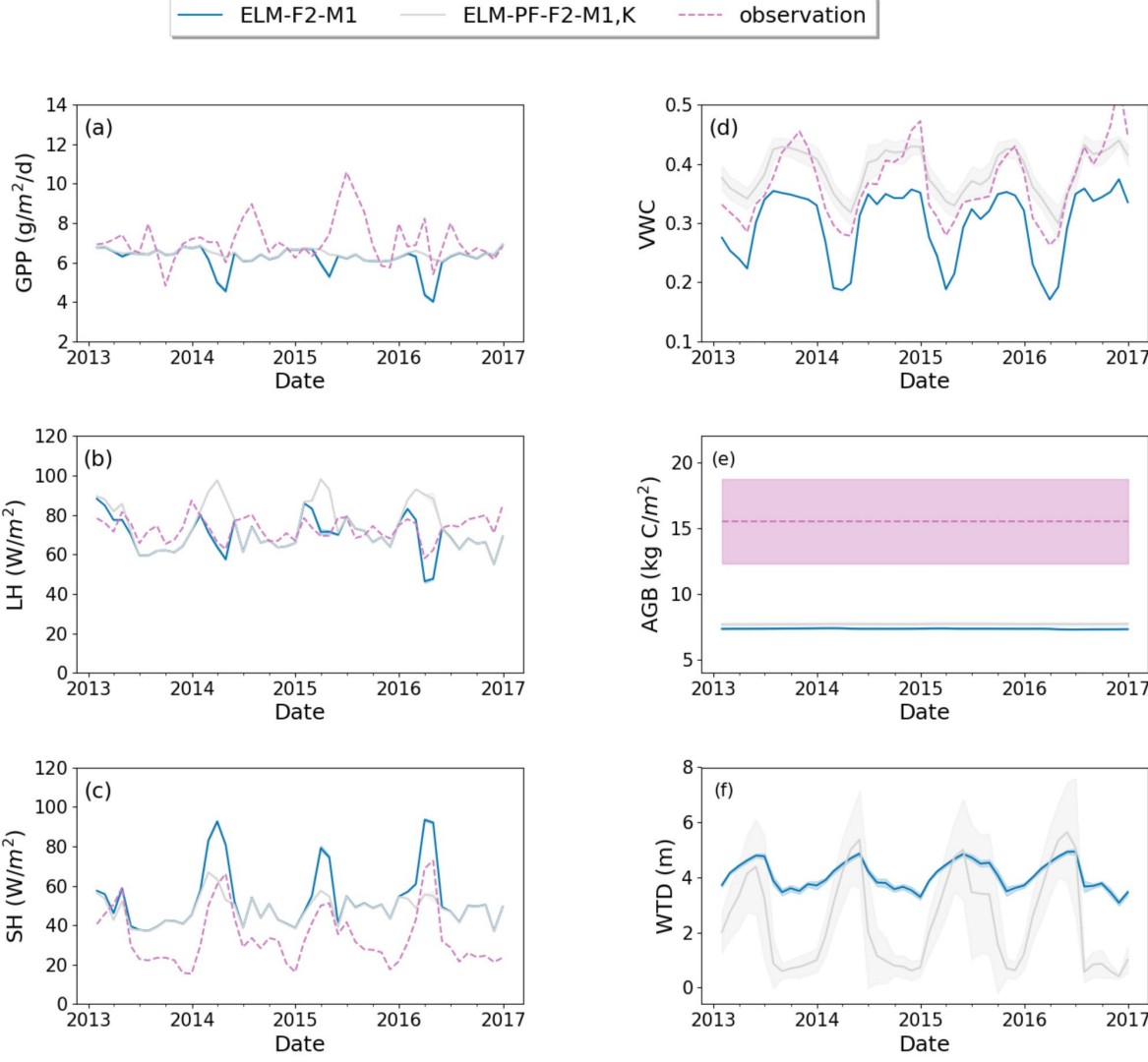

**Figure 5**. Simulated GPP (a), latent heat flux (LH) (b), sensible heat flux (SH) (c), top 15 cm volumetric water content (VWC) (d), aboveground biomass (AGB) (e), and groundwater table depth (WTD) (f) for simulations with default soil property (ELM-F2-M1), and soil property derived from Kupers et al. [2019b] (ELM-PF-F2-M1,K). Dashed line is the observation if available. Solid line is spatial average and shaded area is the standard deviation over the simulation domain.

### 3.3 Impact of water availability on sitewide vegetation structure and mortality

As there is no spatial observation of the WTD at the site, this section is for model comparison only. The simulated AGB decreases nonlinearly with WTD and becomes flat at WTD around 15 m (Fig. 6 a,e) when Parflow is coupled. When hydraulic mortality is triggered (M2 and M3), the slope of the relationship between AGB and WTD (dAGB/dWTD) increases, so WTD plays a larger role in limiting AGB. As AGB does not fluctuate seasonally, the slope

becomes large in the wet season. On the other hand, AGB has a positive relationship with soil moisture content (VWC) (Fig. 6 b,f) and reaches maximum when the soil water content is near saturation. AGB from ELM-PF-F1-M1 is the least sensitive to water table depth because the plant wilting factor (Eq. 12) calculated at the site is much larger than the prescribed threshold of $10^{-6}$, which results in no mortality due to hydraulic failure. ELM-PF-F1-M2 simulates wetter soil

in the dry season compared to ELM-PF-F1-M3 because of hydraulic redistribution simulated by ELM-PF-F1-M2 using FATES-hydro. However, the fast decrease of AGB with WTD using the M2 and M3 functions seems to be unrealistic and requires future exploration. The variability of the normalized AGB across the whole simulation domain considering hydraulic mortality is 0.08, which is comparable to that from the observation, but the variability at the 50 ha plot is still quite

low because of the relatively homogeneous soil hydrology there. Note the water table may be artificially increased near the outer boundary due to the no flow assumption, but a sensitivity experiment using the water level in Gatún Lake at the outer boundaries shows no impact on the conclusion that lower areas are more resilient to water stress and has more biomass.

        AGB from ELM-PF-F1-M2 is smaller compared to that from ELM-PF-F1-M3, especially at

locations where the water table is shallow (WTD < 2 m). That is due to the higher mortality rate triggered by hydraulic failure in ELM-PF-F1-M2 at those locations (Fig. 6c), resulting in less grids with AGB > 4.5 kg C/m$^2$ (Fig. S2). Hydraulic mortality rates from ELM-PF-F1-M2 are much lower than those from ELM-PF-F1-M3 in the dry season (Fig. 6 c), even though at the plateaus WTD is simulated greater than 15 m for both models. The high hydraulic mortality rates

within WTD between 0 to 5 m for ELM-PF-F1-M2 are associated with trees of diameter at breast height (DBH) greater than 16 cm. Mortality from hydraulic failure outcompetes mortality by carbon starvation for ELM-PF-F1-M3, and there is almost no carbon starvation related mortality in both wet and dry seasons when WTD is greater than 7.5 m  (Fig. 6 d,h) because of the reduced maintenance and turnover requirements with fewer trees with DBH between 1 cm and 5 cm. For

ELM-PF-F1-M2, mortality related to carbon starvation and hydraulic failure co-occurs with

similar magnitude in the dry season. In the wet season, there is almost no mortality related to hydraulic failure except for tall trees with DBH > 16 cm dominant in regions of shallow water table depth. Tall trees are hydraulically more vulnerable than short trees because of their more negative stem water potentials due to longer hydraulic path length [*McDowell et al.*, 2002].

Carbon starvation mortality consistently occurs during the dry and wet season when water table depth is greater than 15 m. Carbon starvation mortality rates for ELM-PF-F1-M2 and ELM-PF-F1-M3 decrease with WTD between 0-7.5 m as hydraulic mortality rates increase.

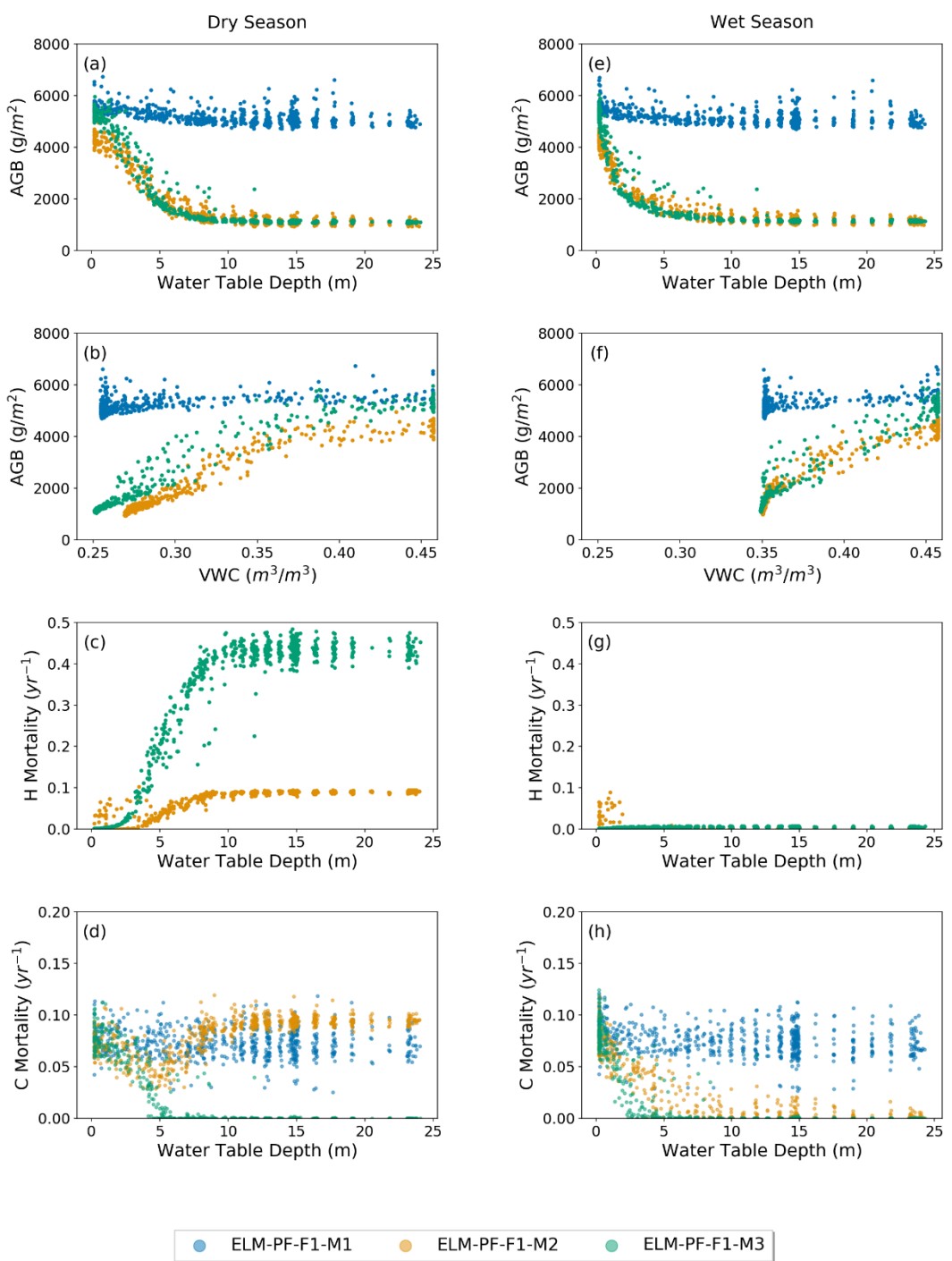

**Figure 6.** The blue, orange, and green circle represent results from ELM-PF-F1-M1, ELM-PF-F1-M2, and ELM-PF-F1-M3, respectively. Simulated aboveground biomass (AGB) with respect to

groundwater table depth, WTD (a,e) and top 1 m soil water content, VWC (b,f), and simulated hydraulic mortality (c,g), carbon starvation mortality (d,h) with respect to WTD the dry season (a-d) and wet season (e-h).

## 3.4 Environmental and physical controls on the simulated results

The RF models have shown good performance. They can explain 90% and more of the variance (VAR$_{ex}$ in Table 2) in AGB and WTD for both the training data and the unseen test data, suggesting the predictors selected are sufficient to explain AGB and WTD. They perform better for AGB than for WTD with mean absolute percentage error (MAPE) less than 10% as opposed to 30% for WTD (Table 2). All explanatory variables used as predictors in the RF models can capture portions of the variability of the simulated AGB and WTD, but the relative importance of the predictors is different for the different ELM-PF models (Fig. 7). Among the three predictors, convexity is most important for describing the spatial variabilities of AGB simulated from ELM-PF-F1-M1. The variable importance for AGB is similar between ELM-PF-F1-M2 and ELM-PF-F1-M3, with slope showing the highest importance (Fig. 7a). For WTD, the variable importance for ELM-PF-F1-M1 and ELM-PF-F1-M3 are comparable (Fig. 7b) as there is no feedback to soil water from plant roots in either model. But convexity and slope play more important roles than DEM in simulating WTD for all models (Fig. 7b) as slope influences water movement [*Famiglietti et al.*, 1998; *Moore et al.*, 1988; *Nyberg*, 1996] and convexity is associated with distance to drainage channels, i.e., whether an area in a hydrologic network is a local depression (valley, swamp) or peak (hilltop, ridge) [*Detto et al.*, 2013].

Introducing the vertically averaged volumetric water content (VWC), for example, from the first month of the various simulations as an additional predictor, the RF models have lower AGB error (column AGB$_{RF2}$ vs column AGB$_{RF1}$ in Table 2) and explain more variance in both the training and test data for all models, and VWC becomes the most important feature for ELM-PF-F1-M2 and ELM-PF-F1-M3 as hydraulic mortality is tied to soil water status. Similar accuracy of the RF models can be achieved if WTD is introduced as additional predictor. These results highlight the importance of representing the interactions between the dynamic physical processes and the static topographic attributes in controlling AGB.


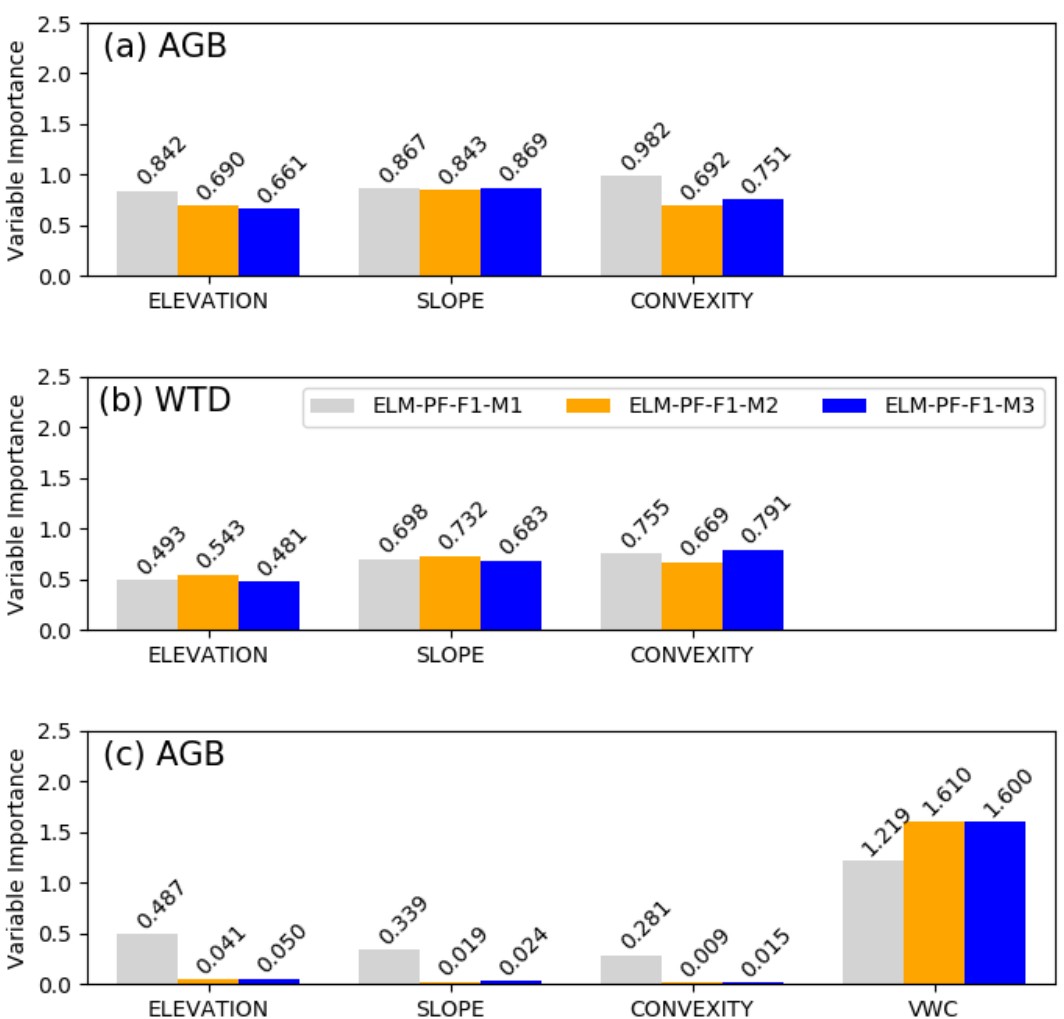

**Figure 7.** Variable importance for the explanatory variables (x-axis) included in the random forest model for the sitewide simulated AGB (a), and WTD (b) as response variables using elevation, slope, and convexity as explanatory variables, and for the simulated AGB (c) using VWC as additional explanatory variables. The number on top of each bar is the importance value.

**Table 2**. Random Forest Model Performance on the simulated above ground biomass (AGB) and water table depth (WTD) from the site wide and 50-ha locations, respectively. Model performance

 is quantified by mean absolute percentage error (MAPE (%)) and percent of variance explained (VAR$_{ex}$ (%)). The paired data separated by "/" in each column are metrics for training data (left) and unseen test data (right). Subscript RF1 indicates the random forest models using topographic features while subscript RF2 indicates model using simulated soil moisture as predictor in addition to the predictors used in RF1 models.

| | Case | Sitewide | | | 50-ha | | |
|---|---|---|---|---|---|---|---|
| | | AGB$_{RF1}$ | AGB$_{RF2}$ | WTD | AGB$_{RF1}$ | AGB$_{RF2}$ | WTD |
| MAPE (%) | ELM-PF-F1-M1 | 0.34/0.38 | 0.27/0.3 | 28.4/32.3 | 0.4/0.5 | 0.23/0.5 | 13.4/13.7 |
| | ELM-PF-F1-M2 | 4.4/4.9 | 4.05/4.56 | 31.2/35.5 | 5.1/6.0 | 2.7/4.8 | 11.4/12.6 |
| | ELM-PF-F1-M3 | 5.2/5.9 | 4.85/5.42 | 27.7/31.6 | 6.4/7.4 | 1.1/2.6 | 11.5/12.6 |
| VAR$_{ex}$ (%) | ELM-PF-F1-M1 | 98.5/98.1 | 99.7/99.6 | 92.7/91.4 | 97.8/96.6 | 98.8/95.1 | 78.1/79.4 |
| | ELM-PF-F1-M2 | 99.1/98.9 | 99.7/99.6 | 91.7/89.8 | 81.3/77.8 | 93.4/79.8 | 84.4/81.4 |
| | ELM-PF-F1-M3 | 99.1/98.9 | 99.7/99.6 | 93.0/91.8 | 38.3/27.0 | 96.7/88.1 | 83.8/80.6 |


Using the same approach as described above for the domain-wide simulations, we also develop RF regression models to identify the important explanatory variables that can describe the simulated AGB and WTD and the observed AGB and VWC at the 50-ha plot in 2015. The RF model for the observation is at 5 m resolution based on the DEM from the BCI census
database. We first analyze the results from the RF models developed based on model simulations at the 50-ha plot. All variables have almost the same level of importance describing the WTD results for ELM-PF-F1-M2 and ELM-PF-F1-M3 (Fig. 8b), but slope is more important than DEM and convexity for ELM-PF-F1-M1. For AGB, the variable importance shows larger differences across the predictor variables and the models. For example, convexity is more
important in describing AGB than DEM and slope for ELM-PF-F1-M2 while slope is much

more important than DEM and convexity in describing ABG for ELM-PF-F1-M1 (Fig. 8a). The accuracy of the RF model for AGB simulated by ELM-PF-F1-M3 is the lowest with high MAPE (6.4%) and the RF model is not able to capture the underlying spatial variability of the data, explaining less than 40% of the variance (Table 2). Hence the predictor variables are

uninformative with respect to the simulated AGB at the 50-ha plot as the plot is fairly homogeneous topographically. When adding VWC as an explanatory variable, it is the most important variable to describe the AGB simulated by ELM-PF-F1-M3 (Fig. 8c) as the hydraulic mortality is a linear function of VWC. It can explain more than 80% of the variance (Table 2). VWC is also important for ELM-PF-F1-M2 to describe the simulated AGB because plant water

is linked to soil water. The accuracies of AGB are all improved when VWC is added as predictor (Table 2). When there is almost no hydraulic mortality (ELM-PF-F1-M1), slope is the dominant driver for the simulated AGB and WTD (Fig. 8a,c).

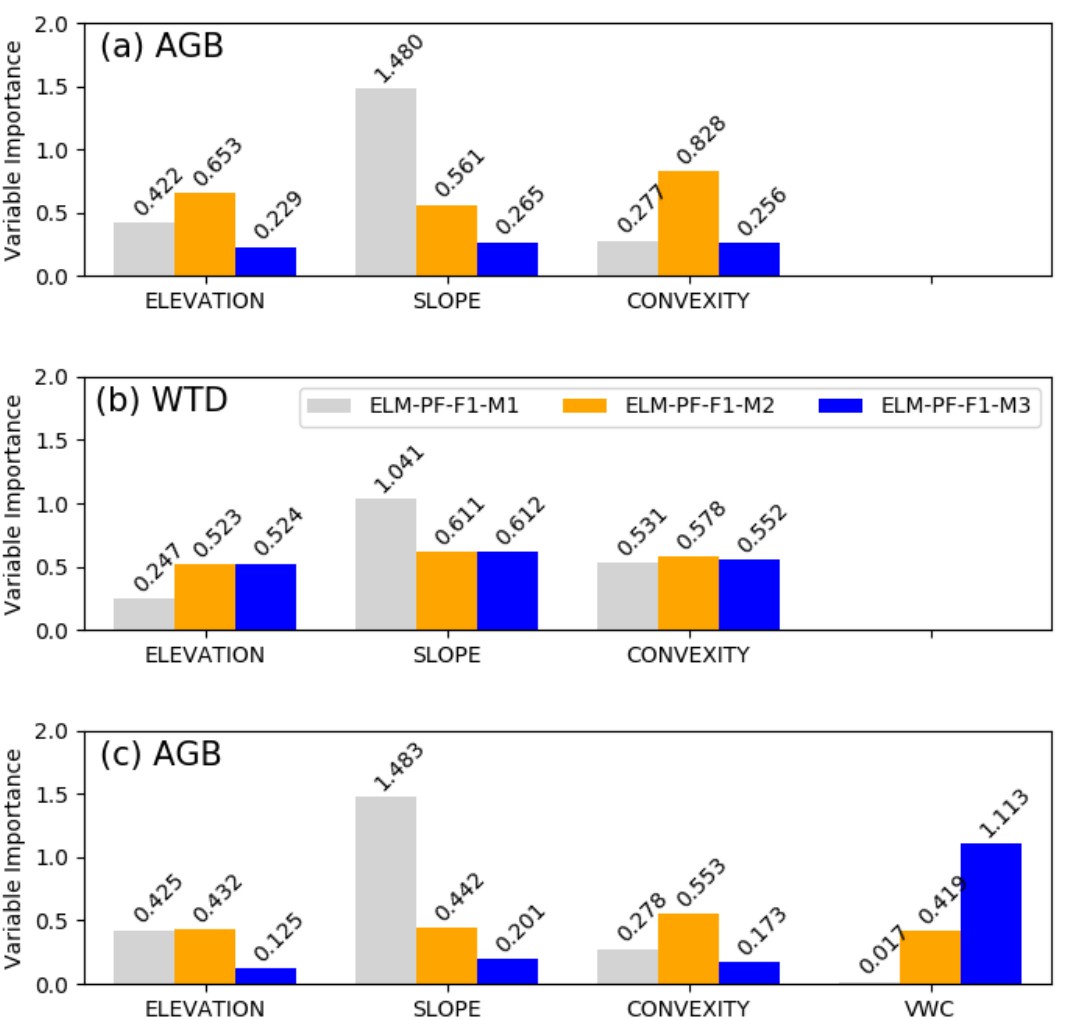

**Figure 8.** Variable importance for the explanatory variables (x-axis) included in the random forest model for the simulated AGB (a) and WTD (b) as response variables using elevation, slope, and convexity as explanatory variables, and for the simulated AGB (c) using VWC as additional explanatory variables in the 50-ha plot. The number on top of each bar is the importance value.

Compared to the RF regression model for the simulated AGB and VWC, explanatory variables including DEM, slope, and convexity can also well describe the observed VWC with

57.5% variance explained for the training data and 46.8% for the test data and MAPEs are 4.0% and 4.4% for the training and test data, respectively. DEM and slope have a slightly higher importance compared to convexity for the observed VWC (not shown). However, the RF model of the observed AGB using the topographic features and the observed VWC as explanatory variables can only master the training data. Even though it finds that slope is an important driving factor in the training data of the observed AGB, as supported by Mascaro et al. [2011] using the multiple regression method to examine controls over AGB derived from airborne Light Detection and Ranging (LiDAR) at BCI, it cannot master the test data (negative explained variance). Thus, the RF model based on the observed AGB is not able to generalize well. All the predictor variables including the observed VWC besides DEM, slope, and convexity are uninformative for the spatial variability of the observed AGB. This suggests that the data is sparse and/or the observed AGB may depend on other factors such as soil heterogeneity and nutrient availability.

## 4.  Discussions and conclusions

There are many external and internal factors controlling ecosystem functioning, one of which is plant water availability. We developed a model to incorporate 3D subsurface modeling in Earth system in consideration of the role of hillslope on water availability and vegetation dynamics under water stress conditions. We applied the model to BCI where sustained water stress on canopy trees has been reported in the past.

Our domain-wide simulations using ELM and the coupled ELM and ParFlow showed WTD can differ significantly from the wet lowland (shallow WTD) to the dry highland (deep WTD) when lateral flow is introduced by coupling ELM with ParFlow. The large difference in WTD affects soil water availability along the topographic gradient and consequently causes large spatial variability in the energy flux partitioning and GPP compared to ELM when soil hydrology is represented by vertical one-dimensional flow. As summarized in the review paper by Fan et al. [2019] and references therein, this spatial variability in energy and water associated with topography can fundamentally organize the vegetation structure, energy, and biogeochemical fluxes across the landscape under water and energy limiting conditions.

Coupled to the subgrid vegetation dynamics model FATES, we found higher AGB in the wet areas compared to dry areas in the domain-wide simulations. AGB decreases nonlinearly

with increasing WTD when WTD is less than 10 m, but the relationship asymptotes beyond WTD of 10 m. Unlike WTD, AGB increases almost linearly with increasing VWC over a wide range of VWC values. When hydraulic failure occurs under water limiting conditions, the

biomass difference along the topographic gradient can further increase. Consistent with the higher VWC during the wet season compared to the dry season, mortality rates from hydraulic failure are very low in the wet season and model differences become minimal. Hydraulic failure represented by different methods can affect the mortality rate induced by carbon starvation. For example, using the approach in Eq. (14) to represent drought mortality rate as a linear function of

soil water potential, there is essentially no carbon starvation for areas with WTD deeper than 10 m.

Even though soil water gradients have been identified as an important determinant of tropical forest structure and functioning [*Miron et al.*, 2021; *Terra et al.*, 2018], unlike for the modeling results, we were not able to find similar relationship of the corresponding observed features with

observed AGB. More specifically, the results of our RF model based on observations reveal that topographic attributes and soil water availability cannot explain the spatial distribution of aboveground biomass observed at the 50-ha plot located in the highland of BCI, with relatively homogeneous surface conditions. While the lack of relationships in the observations may reflect the limited data coverage, it also suggests other factors may potentially play an important role in

driving the spatial variability of the observed AGB. Furthermore, differences in the explanatory power of the topographic attributes and soil water status for the simulated AGB and the lack thereof for the observed AGB suggest that factors that drive the spatial variability of the observed AGB are likely absent or not well represented by the coupled model. The water table at the plot is close to the surface with several springs on the slopes [*Becker et al.*, 1988; *Harms et*

*al.*, 2001] and there were considerable and non-systematic variation in soil saturated hydraulic conductivity [*Kinner and Stallard*, 2004] that could generate preferential flow paths. These observed features, which are not accounted for in our model, could limit the ability of the coupled model in reproducing the observations, even if more systematic efforts were devoted to calibrate the model parameters. And they should be explored in future studies. Other factors

currently not accounted for by the model include spatial biodiversity of functional traits, toxic metals, soil nutrients, and liana (woody vines) abundance, which have all been found to influence the tree AGB at BCI [*Ingwell et al.*, 2010; *Ledo et al.*, 2016; *Schnitzer and Bongers*,

2011; *Schnitzer et al.*, 2005; *Zemunik et al.*, 2018]. Also not accounted for by the model is the negative relationship between the wood density and tree mortality rate at BCI found in McDowell et al. [2018] using data from Wright et al. [2010]. Including spatially variable tree mortality based on a negative relationship between mortality and wood density may substantially improve model representation of vegetation carbon as indicated by the modeling study in Hancock et al. [2022]. Local heterogeneity of plant functional composition and soil resources should be considered in future models [*Hofhansl et al.*, 2020].

Accurate estimation of spatial AGB and its dynamics is important for global carbon cycle and climate mitigation. Lateral flow that has a strong influence on soil water gradient is often missing in ecosystem modeling. Using a coupled land model, 3D integrated hydrologic model, and ecosystem dynamics model to simulate the carbon stock distribution at BCI, we found the simulated AGB is strongly influenced by topographic attributes and/or soil water availability at larger scale if hydraulic failure is triggered. However, prescribing spatially homogeneous soil properties and plant traits, the coupled model cannot explain the observed larger variability in AGB in the highland where WTD variations are likely very small. We also found drought mortality as a function of hillslope soil moisture (Eq. 14) or due to plant hydrodynamics (Eq. 12) can contribute to the large spatial variability in AGB. These two hydraulic failure models are easily introduced in our coupled model without having to empirically parameterize the hydrology model. However, these two models have different effect on carbon starvation mortality. Data needs to be collected to support the findings in this study. For example, soil moisture, WTD, AGB, and plant traits (e.g., wood density) across hydrologic gradient (from low land to high land). It is necessary to have a better quantification of the soil texture and related hydraulic properties as the distribution of biomass is the combined result of plant traits, soil properties, climatic and groundwater conditions [Costa et al., 2022]. AGB can be influenced by soil texture which directly affects the time interval between precipitation inputs and groundwater recharge [Sousa et al. 2022] and the capillary fringe above the water table that supplies water to the rooting zone [Costa et al. 2022]. For example, results in Sousa et al. [2022] suggest a contribution of clayey texture in increasing AGB in dry climates with a shallow water table. Spatial heterogeneity is lacking in many forest dynamics models [*Busing and Mailly*, 2004].

Future modeling research should also account for spatial heterogeneity of soil resource (i.e., water and nutrients) and plant functional traits (e.g., mortality, growth, rooting depth etc.), as

well as anthropogenic factors (habitat loss due to deforestation, degradation, and fragmentation

[Miranda et al., 2017]) on the structure of plant communities. As a demonstration, only two plant functional types were considered in this study. When water stress is considered, the negative response of AGB with WTD simulated by the model is supported by previous studies (e.g., Esteban et al., 2021) that species associated with deep water tables had decreased growth and increased mortality compared to those associated with shallow water table depth during severe

drought. However, the two hypothetical hydraulic failure models (M2 and M3) result in a strong positive relationship between mortality with the soil water stress, driving unrealistic response of AGB to increasing WTD. In reality, the resistance of trees to water stress also depends on the severity of droughts, plant traits, and environmental conditions [Costa et al., 2022; and references therein]. For example, previous studies found that hydraulically-vulnerable trees can

delay dehydration by accessing deep water during droughts in BCI [Chitra-Tarak et al., 2021]. How plant traits and PFT composition will impact these rates should be a key consideration in advancing coupled modeling in the future. The coupled ELM-PF-FATES will be applied to other tropical forest regions where lateral flow and groundwater dynamics may play different role in water available to plants to further elucidate carbon-hydrology interactions and plant

response to drought.

Using a three-dimensional model in current Earth system models that are typically run at ~100 km grid resolution may yield inaccurate results or have no significant on vegetation dynamics. A reasonable grid resolution for groundwater flow simulation is around 1 km (Xie et al. 2020 and references therein). Moving from 100 km to 1 km resolution for global scale

vegetation dynamics simulation coupled with a 3D integrated hydrologic model is computationally challenging, but it may be a realistic goal with advances of computation power and architecture in the future. The model in this study provides opportunities for improving hydrological, ecological, and meteorological predictions of Earth system models.

*Code and data Availability*. The coupled code is available at

https://doi.org/10.5281/zenodo.6595795. The census data for the BCI plot are publicly available at http://dx.doi.org/10.5479/data.bci.20130603. Use of the data has been agreed upon with the principal investigators of the plot: Stephen Hubbell, Richard Condit and Robin Foster. Other observational data are available at http://doi.org/10.5281/zenodo.3752127.

*Author contributions.* YF, GB developed the code. YF set up the model, performed simulations and prepared the figures, YC prepared the model parameters. RL, CK, MD, NM, HM, JW, and JC contributed to discussion, writing and editing.

*Competing interests.* The authors declare that they have no conflict of interest.

## Acknowledgments

This work was supported by the U.S. Department of Energy Office of Biological and Environmental Research as part of the Terrestrial Ecosystem Systems program through the Next Generation Ecosystem Experiment (NGEE) Tropics project. The Center for Tropical Forest Science - Forest Global Earth Observatory (CTFS-ForestGEO) supported the flux tower research. MD was supported by the Carbon Mitigation Initiative at Princeton University and NSF grant 2017804. The BCI forest dynamics research project was founded by S. P. Hubbell and R. B. Foster, and is now managed by R. Condit, S. Lao and R. Perez under the Center for Tropical Forest Science and the Smithsonian Tropical Research in Panama.

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
