# Peer review of "Modeling the topographic influence on aboveground biomass using a coupled model of hillslope hydrology and ecosystem dynamics"

_Geoscientific Model Development, 2022_

## Author Comment (AC1)

**Response to RC1**

We thank the referee for the constructive comments and thorough review for the improvement of our manuscript.

Hydrological mortality influences water availability and biomass. In this manuscript, the authors present the results of a study designed to incorporate the ELM, FATES, and Farflow to show the importance of lateral subsurface in above-ground biomass. The WTD, AGB, and Bowen ratio could be varied after using the three-dimensional hydrology model. This paper is valuable because it is the first to apply the three-dimensional hydrology model to the Earth System model. This work could be helpful for the researchers using ELM, FATES (or ED2). However, model explanation, design of experiments, and the results need more detail, and analysis is ambiguous in several places.

Thanks for the positive comments. Please see our point-by-point responses below.

Major comments

1. In section 2.1.3, while explanations of different hydraulic failure models are well described, the explanation of the biomass equation is insufficient. Clarify it (Equations calculating the above-ground biomass should be needed in this chapter).

   Above-ground biomass calculation in FATES is added and described below.

   FATES uses allometric relationships with stem diameter ($D$) to determine tree height ($h$) and crown area ($C$). There are five model options for tree height in FATES. In this study, we used a power function described in Obrien et al. (1995) for FATES input file F1 in the main text:

   $$h = 10^{(log10(D_*) \cdot a + b)} \tag{R1}$$

   $$D_* = \min{(D, D_{max})} \tag{R2}$$

   and a Michaelis–Menten form in Martinez Cano et al. (2019) for FATES input file F2:

   $$h = \frac{cD_*^d}{k + D_*^d} \tag{R3}$$

The allometry function for crown area is

$$C = \begin{cases} fD^g & D < D_{max} \\ fD_{max}^g & D \geq D_{max} \end{cases} \tag{R4}$$

where $a, b, c, d, k, f,$ and $g$ are allometric parameters, $D_{max}$ is diameter of plant where max height occurs.

Target biomass of leaf, structure, stem, fine root, seed, and storage are also calculated using allometry functions in FATES (Koven et al., 2020). Target biomass of fine root and storage are assumed to be linearly proportional to the target leaf biomass, and the target structure biomass is linearly proportional to the target sapwood biomass.

A power law allometric model is used for the target leaf biomass (L):

$$L = mD_*^g \tag{R5}$$

where $m$ and $g$ are allometric parameters, and $g$ is the same as in Eq. R3.

FATES has three allometry function options to calculate target stem aboveground biomass ($C_{agb}$), we used the functional form in Saldarriaga et al. (1998) in F1:

$$C_{agb} = f_{agb}p_1h^{p_2}D^{p_3}\rho^{p_4} \tag{R6}$$

and a functional form in Chave et al. (2014) in F2:

$$C_{agb} = \frac{1}{c2b}p_1(\rho D^2 h)^{p_2} \tag{R7}$$

where $f_{agb}$ is the fraction of stem above ground, $p_1, p_2, p_3,$ and $p_4$ are allometry parameters, $c2b$ is carbon to biomass ratio, $\rho$ is the plant wood density.

Once tissue turnover and storage carbon demands are met, FATES uses a constant fraction of net primary production for seed production. Total aboveground biomass (AGB) reported in the study is the sum of leaf biomass, aboveground stem biomass and seed biomass.

2. The meteorological data which are employed in the model are from tower measurements. So, the same meteorological forcing is used in all grids? Is it a reasonable method? Moreover, which variables are used in this study (for example, temperature, precipitation, radiation).

Note that the model can use spatial forcing, but we don't have the forcing data at the resolution of our model at BCI. We used the same meteorological forcing in all grids. It is reasonable for this site as the rain measured at the clearing near the Lutz catchment and the AVA tower (at the plateau of the 50 ha plot, ~1.25 km from the Lutz) agree pretty well on a monthly scale (Figure R1).  Precipitation, air temperature, relative humidity, wind speed, surface pressure at the tower are used in this study.

[Figure]

Figure R1. Comparison of precipitation at the AVA tower and at a clearing near the Lutz catchment.

This sentence now reads:

The model is driven by the same atmospheric forcing (i.e., precipitation, air temperature, relative humidity, wind speed, and surface pressure) for 2003-2016 measured at a meteorological tower near the Lutz catchment at BCI [*Faybishenko et al.,* 2018] in all grids due to the lack of spatial forcing. Comparison of the precipitation at the tower and a clearing near the Lutz catchment shows good agreement supporting the use of the same atmospheric forcing for all grids of the model.

3.  Before starting the experiments, parameter tunning of the FATES model should be done because the FATES model has large uncertainties yet. The above-ground biomass in ELM-F1-M1, ELM-F1-M2, which are basic models (figure4 e) is quite underestimated because the parameter tunning was not done. After the parameter tunning was done, the results all could be different. (The effect of the 3-dimensional model on biomass may be reduced or increased.)

    The FATES parameters were selected from previous ensemble simulations for tropical forests which produced results close to the observations. We agree the results could be different after parameter tuning, but they will not affect our conclusion that water table dynamics can play large role when hydraulic failure is triggered. We made clarification in section 2.4:

    Two PFTs representing early successional and late successional species are simulated at the same time in competition with each other using two input files of plant traits selected from previous ensemble simulations that best matched observations for tropical forests [Chen et al. 2022, Huang et al. 2020]. Further parameter tuning is out of the scope of this work.

4.  With the 3-dimensional model, the AGB and GPP were simulated bigger at lowland than at highland. Do you have observation or reference? Does the fact that the simulated wetter soil in lowland using a three-dimensional model show the improving model performance or just model comparison?

    Unfortunately, we don't have spatial observations that cover both lowland and highland to validate the simulated AGB and GPP. The difference at lowland and highland is model comparison. It is clarified as:

    Note this is based on model comparisons. Spatial observations at those locations are needed to validate the model but such observations are not currently available.

5. In section 3.2, while the results of ELM-PF-FL-M1 are well written, the explanation of ELM-PF-F2-M1 was quietly insufficient. Moreover, what is difference between F1 and F2? This explanation could be added in section 2.4.

   Thanks for the suggestion. We added explanation of ELM-PF-F2-M1. Please see our response to the following Comment 6. The main difference between F1 and F2 are the allometric models described in Comment 1 above. F2 also has lower carbon starvation mortality rate. These two input files are now included in the Supplement. The following descriptions are added in section 2.4:

   The allometric models for tree height and target stem aboveground biomass in F1 are defined in Eqs. 4 and 9, respectively, and those in F2 are defined in Eqs. 6 and 10, respectively.

   The complete parameters for F1 and F2 are included in the Supplement.

6. With F2, the AGB increases, but GPP decreases. In general, the AGB is positive correlated with GPP, but it is not in this case. This could be discussed with a trait of plants. (Line 476). And it would be better to think about it the direct impact of plant trait on AGB and the indirect impact (plant trait -> WTD -> AGB)

   As the soil moisture simulated using F1 and F2 are close, GPP is mainly affected by growth allometry while AGB is the result of both growth and mortality. From Fig. R2 we can see that using F2 results in larger growth rates and significantly lower mortality rates, thus increasing AGB for F2 compared to F1. However, simulation with F2 results in much lower exposed leaf area index, thus lower GPP compared to that with F1. If we use the same input file, increasing only $V_{cmax}$ will increase both AGB and GPP as shown in Fig. 5. The following are added in the manuscript in response to Comments 5 and 6. Fig. S1 below refers to Fig. R2 in this response.

   The spatial standard deviations (STDs) of the aforementioned-variables of interest for simulation ELM-PF-F2-M1 are slightly smaller than for ELM-PF-F1-M1. The difference in STD between ELM-PF-F2-M1 and ELM-PF-F1-M1 is larger for VWC, LH, and SH compared to GPP, AGB and WTD. With this plant traits F2, AGB increases by 47.5% and GPP decreases by 19% on average (Fig. 4). As the soil moisture (VWC) simulated using ELM-PF-F1-M1 and ELM-PF-F2-M1 are close , GPP is mainly affected by growth allometry while AGB is the result of both growth and mortality. Using plant traits F2 results in larger growth rates and significantly lower mortality rates (Fig. S1), thus increasing AGB for F2 compared to F1. However, simulation with F2 results in much lower exposed leaf area index, thus lower GPP compared to that with F1.

[Figure]

**Figure R2**. Growth (increment of diameter at breadth height, DDBH) (a) vs. mortality (b), and exposed leaf area index (c) for ELM-PF-F1-M1 and ELM-PF-F2-M1. Solid line is spatial average and shaded area is the standard deviation over the simulation domain.

7. In-Line 479~484, I was confused that the experiment with adjusting the VCMAX was shown suddenly. There is no experiment in Table 1 and in Section 2.4. And why did you multiply 1.9? Is there any reference?

   As it was pointed out in Comment 3, our untuned parameters underestimated AGB. This experiment was introduced to show that tuning parameters can improve the simulated AGB, but the AGB variability is still too small compared to the observation. The factor of 1.9 is determined through trial and error, and

there is no reference. We removed the experiment from the text and Fig. 5 to avoid confusion. It now reads

AGB can be further increased by parameter tuning for better agreement with observations, but we don't expect it to significantly change the AGB variability.

8. In section 3.3. it is very interesting about the response of AGB, VWC, Water table depth, H mortality and C mortality to M1, M2 and M3 model. In Line 514, you mentioned that M2 simulates wetter soil in the dry season compared to M3. But it seems likely that simulated AGB in M3 is more than M2 (Figure 6 (a), (b)). Could you add the results of AGB in M2 and M3 in dry seasons and discuss that why is AGB in M3 is more than M2? To my knowledge, the vegetation in wetter soil makes more carbon, especially in dry seasons.

Thanks for the comment. The higher AGB in M3 is mainly at locations near the shallow water table where M3 simulates lower hydraulic mortality. We added the results and discussion shown below. Fig. S2 in the following refers to Fig. R3 in this response.

AGB from ELM-PF-F1-M2 is smaller compared to that from ELM-PF-F1-M3, especially at locations where the water table is shallow (WTD < 2 m). That is due to the higher mortality rate triggered by hydraulic failure in ELM-PF-F1-M2 at

those locations (Fig. 6c), resulting in fewer grids with AGB > 4.5 kg C/m² (Fig. S2).

[Figure]

**Figure R3**. Histogram of simulated AGB for ELM-PF-F1-M2 and ELM-PF-F1-M3

9. In section 3.4, the description of variable importance of AGB, WTD was written. How did you calculate the importance? It should be added in section 2.5.

We added the following in section 2.5:

To calculate the permutation importance, a reference score (prediction error) for a trained regression model *m* is first calculated. Each feature *j* (a column) in the training or testing dataset is randomly shuffled to generate a corrupted dataset and the score of the model *m* on the corrupted dataset is calculated. The shuffling and corrupted dataset score computation are repeated multiple times. The importance of feature *j* is computed as the difference between the reference score and the arithmetic mean of the scores of the model *m* on the

corrupted datasets. This is documented in https://scikit-learn.org/stable/about.html#citing-scikit-learn.

10. It is necessary to first show the performance of how well AGB and WTD are predicted by the random forest method. It is necessary to show the importance after presenting the results and discussing them. Moreover, why did you select elevation, slope, convexity, and vwc using the random forest? Is this sufficient to explain the importance of AGB and WTD?

Thanks for the comment. We adjusted the order of results and the importance and it now reads:

The RF models have shown good performance. They can explain 90% and more of the variance ($VAR_{ex}$ in Table 2) in AGB and WTD for both the training data and the unseen test data, suggesting the predictors selected are sufficient to explain AGB and WTD. They perform better for AGB than for WTD with mean absolute percentage error (MAPE) less than 10% as opposed to 30% for WTD (Table 2). All explanatory variables used as predictors in the RF models can capture portions of the variability of the simulated AGB and WTD, but the relative importance of the predictors is different for the different ELM-PF models (Fig. 7).

The reason we selected elevation, slope, convexity, and vwc is because they have previously been shown to affect AGB and WTD, and the reason was provided at the beginning of section 2.5 in the original submission. Our selection of the predictor variables is supported by the results of our random forest models showing that those predictors are sufficient to explain AGB and WTD.

11. This paper has novelty because a three-dimensional model is incorporated with Earth system modeling. In this paper, the domain is very small, while the domain and spatial resolution of research using the Earth system model are still large. Using a three-dimensional model may have inaccurate results or no significant impact with large resolution. It would be good to add a discussion on using a three-dimensional model in a global scale.

Thanks for the positive comment and suggestion. We added the following discussion:

Using a three-dimensional model in current Earth system models typically run at ~100 km grid resolution may yield inaccurate results or have no significant impact on vegetation dynamics. A reasonable grid resolution for groundwater flow simulation is around 1 km (Xie et al. 2020 and references therein). Moving from 100 km to 1 km resolution for global scale vegetation dynamics simulation

coupled with a 3D integrated hydrologic model is computationally challenging at present, but it may be a realistic goal with advances of computation power and architecture in the future. The model in this study provides opportunities for improving hydrological, ecological, and meteorological predictions of Earth system models.

Minor comments

Contracted words are inappropriate in a science paper.

Line 103 during1983 -> during 1983

Thanks for the correction.

Line 198 The words of leaf area index are repeated

Thanks. Deleted.

Line 293~295 The sentence about ET could be deleted

The sentence is rewritten as to describe the source of the observations. It now reads

Observed evapotranspiration (ET), gross primary production (GPP), sensible heat flux (SH), and latent heat flux (LH) at the site were obtained from an eddy-covariance system installed in July 2012 on the AVA tower.

Line 318~319 and 321~322 These sentences should be rearranged (-> in section 2.3)

Thanks. They are moved to section 2.3.

Line 333 2)->1)

Thanks. Corrected.

Line 349~357 The explanations about spin up and experimental design are quietly hard to understanding. The case 3 was run 200 years and after that additionality runs for 16 years was done due to the comparing case 6 and case 7. It would be nice to show this in Table 1. And did you run the CASE 5 after case 4?

Yes, Case 5 was run after Case 4. Table 1 is revised as in the following:

**Table 1**. Definition of model experiments with ELM, PF, F, and M denoting E3SM land model, ParFlow, different parameters for plant traits, and different mortality models, respectively. "K" in experiment name of Case 5 indicates soil property derived from Kupers et al. [2019b] is used. Extra 16 years of simulation were conducted for four experiments.

| Cases | Model Experiments | Plant Traits | Soil Property | ParFlow | Drought Mortality Model | Extra simulation years for model comparison |
|-------|-------------------|--------------|---------------|---------|-------------------------|---------------------------------------------|
| 1 | ELM-F1-M1 | F1 | S1 | No | Eq. (11) | 0 |
| 2 | ELM-F2-M1 | F2 | S1 | No | Eq. (11) | 0 |
| 3 | ELM-PF-F1-M1 | F1 | S1 | Yes | Eq. (11) | 16 from Case 3 |
| 4 | ELM-PF-F2-M1 | F2 | S1 | Yes | Eq. (11) | 0 |
| 5 | ELM-PF-F2-M1,K | F2 | S2 | Yes | Eq. (11) | 16 from Case 4 |
| 6 | ELM-PF-F1-M2 | F1 | S1 | Yes | Eq. (13) | 16 from Case 3 |
| 7 | ELM-PF-F1-M3 | F1 | S1 | Yes | Eq. (14) | 16 form Case 3 |

VWC is already shown in Figure 6, but there is no description of VWC in section 3.3. The first introduction of VWC is in section 3.4

Thanks! We added the introduction of VWC after "soil moisture content" in section 3.3.

Line 610 adds the discussion why the results of test data was not good.

We discussed that "the data is sparse and/or the observed AGB may depend on other factors such as soil heterogeneity and nutrient availability" at the end of the paragraph in the original submission.

References:

Chave, J., Réjou-Méchain, M., Búrquez, A., Chidumayo, E., Colgan, M. S., Delitti, W. B. C., Duque, A., Eid, T., Fearnside, P. M., Goodman, R. C., Henry, M., Martínez-Yrízar, A., Mugasha, W. A., Muller-Landau, H. C., Mencuccini, M., Nelson, B. W., Ngomanda, A., Nogueira, E. M., Ortiz-Malavassi, E., Pélissier, R., Ploton, P., Ryan, C. M., Saldarriaga, J. G., and Vieilledent, G.: Improved allometric models to estimate the aboveground biomass of tropical trees, Glob. Change Biol., 20, 3177–3190, 2014

Martínez Cano, I., Muller-Landau, H. C., Wright, S. J., Bohlman, S. A., and Pacala, S. W.: Tropical tree height and crown allometries for the Barro Colorado Nature Monument, Panama: a comparison of alternative hierarchical models incorporating interspecific variation in relation to life history traits, Biogeosciences, 16, 847–862, https://doi.org/10.5194/bg-16-847-2019, 2019.

O'Brien, S. T., Hubbell, S. P., Spiro, P., Condit, R., and Foster, R. B.: Diameter, Height, Crown, and Age Relationship in Eight Neotropical Tree Species, Ecology, 76, 1926–1939, 1995.

Saldarriaga, J. G., West, D. C., Tharp, M. L., and Uhl, C.: Long-Term Chronosequence of Forest Succession in the Upper Rio Negro of Colombia and Venezuela, J. Ecol., 76, 938–958, 1988.

Xie, Z., Wang, L., Wang, Y., Liu, B., Li, R., Xie, J., et al. (2020). Land surface model CAS-LSM: Model description and evaluation. Journal of Advances in Modeling Earth Systems,12, e2020MS002339. https://doi.org/10.1029/2020MS002339

---

## Author Comment (AC2)

**Response to RC2**

We are very grateful to this referee for the thoughtful critique and suggestions, which we believe have improved the readability of the manuscript.

Review of Modelling the topographic influence on aboveground biomass using a coupled model of hillslope hydrology and ecosystem dynamics by Yilin Fang et al

The study combines hillslope hydrological processes and ecosystem demography within an Earth system model framework. This is done by coupling a land component (ELM) of an earth system model (E3SM) using the FATES model as the vegetation demography component with a 3-D hydrology model (ParFlow). The model is applied and evaluated at BCI, Panama using hidrological and vegetation observations from the study site. The scientific aim is to investigate the influence of topography via hydrological processes on AGB. The paper presents a series of model sensitivities to model structure, plant traits, soil properties and hydraulic failure representations.

Combining 3-D hydrology, ecosystem demography and testing of various drought mortality functions on an ESM framework and testing it at site level is worth of publication in GMD. The work is overall well written and mostly clear. Unfortunately, the site selection was not ideal for testing impacts of hillslope water availability on AGB due to the low elevation at the site which could have been known a priory with the digital elevation model information.

Thanks for the positive comments. We selected this site as it has some relevant observations and meteorological forcing available for our model development and testing. ELM-FATES-ParFlow is being applied to another tropical forest region with ongoing measurements of hillslope flow, groundwater table, and vegetation dynamics.

There are various minor comments are important to improve comprehension of the analysis, discussion and conclusion.

The results and discussion section need to make a clearer differentiation of text referring to predictions and to observations. This is not clear in many parts. I was unable to find the figures and tables that refer analysis related to AGB observations. A suggestion is that sections that are only model sensitivity analysis and do not use the observations need to be separated from the section dealing with observations.

Thanks for the suggestion. We clarified in each section that has only model sensitivity analysis. The following sentences are added in section 3.1 and 3.3, respectively:

This section focuses on model sensitivity analysis as no spatial observations are available for comparison to the model simulations.

As there is no spatial observation of the relationship between AGB and WTD at the site, this section is for model comparison only.

Section 3.2 refers to spatial variability of simulated and observed variables with various model configurations , yet it shows temporal figures (fig 4 and 5). Spatial variability of the observations (ABG for example) is no shown, for other variables I understand that only a time series at a single location is available, but maybe should not refer to spatial variability if the comparison is to a single point observation. Some of the work done in this section is new to the results section, i.e not mentioned in the methods (Vcmax sensitivity)

Thanks for the comment. The spatial variability of the model was used to compare the effect of topography on water availability at different elevations and consequentially on the variables of interest. It can help us infer under what conditions the model can better match the observations. We showed spatial variability of variables when available. The Vcmax experiment was removed from the text and Fig. 5 to avoid confusion and replaced with

AGB can be further increased by parameter tuning for better agreement with observations, but we don't expect it to significantly change the AGB variability.

The discussion on factors that could explain observed AGB variability is succinct and vague, this section needs to elaborate further, for example it should include discussion possible variability of wood density.

Thanks for the suggestion. We added some discussions in section 3.2:

Based on the model results, species competition also cannot explain the observed variance of AGB at the 50-ha plot without accounting for the spatial heterogeneity of soil properties, nutrient availability, plant traits, etc. in the model. For example, wood density can contribute to the observed variability as it is a parameter used to define the allometry function (Eq. 10).

Eq. 10 was added in the text. It is a functional form in Chave et al. (2014):

$$C_{agb} = \frac{1}{c2b} p_1 (\rho D^2 h)^{p_2} \tag{10}$$

where $C_{agb}$ is the target stem aboveground biomass, $p_1$ and $p_2$ are allometry parameters, c2b is carbon to biomass ratio, $\rho$ is the plant wood density, $D$ is stem diameter, and $h$ is tree height.

Reference:

Chave, J., Réjou-Méchain, M., Búrquez, A., Chidumayo, E., Colgan, M. S., Delitti, W. B. C., Duque, A., Eid, T., Fearnside, P. M., Goodman, R. C., Henry, M., Martínez-Yrízar, A., Mugasha, W. A., Muller-Landau, H. C., Mencuccini, M., Nelson, B. W., Ngomanda, A., Nogueira, E. M., Ortiz-Malavassi, E., Pélissier, R., Ploton, P., Ryan, C. M., Saldarriaga, J. G., and Vieilledent, G.: Improved allometric models to estimate the aboveground biomass of tropical trees, Glob. Change Biol., 20, 3177–3190, 2014.

The study concludes that data needs to be collected to support findings of this study but does not elaborate (L700). There is a need to inform the ecological/plant physiology community on what is needed to be able to represent these processes/and or parameters needed in Earth System Models.  L701-L702 are equally vague, authors need to be more specific on what is needed.

We added the following for elaboration:

For example, soil moisture, WTD, AGB, and plant traits across hydrologic gradient (from low land to high land).

Future modeling research should also account for spatial heterogeneity of soil resource (i.e., water and nutrients) and plant functional traits (e.g., mortality, growth, rooting depth etc.), as well as anthropogenic factors (habitat loss due to deforestation, degradation, and fragmentation [Miranda et al., 2017]) on the structure of plant communities.

Reference:
Miranda, A., Altamirano, A., Cayuela, L., Lara, A., Gonzalez, M., 2017. Native forest loss in the Chilean biodiversity hotspot: revealing the evidence. Reg Environ Change 17, 285-297.

Section 3.4 is full of statements that miss a figure or a table supporting the text.

Figure and table are now cited to support the text. Please see the response to the specific comments below.

Specific comments

L73, Clark et al 2015 is missing in the reference list, if this refers to JULES, that might be Clark et al 2011,doi:10.5194/gmd-4-701-2011 which focuses on the carbon cycle, Best et al 2011, doi:10.5194/gmd-4-677-2011 has more focus on the energy balance probably more appropriate. Jules has been used to represent ecosystems along topographic gradients,

See Hsi-Kai Chou, Boris F. Ochoa-Tocachi, Simon Moulds & Wouter Buytaert (2022): Parameterizing the JULES land surface model for different land covers in the tropical Andes, Hydrological Sciences Journal, DOI: 10.1080/02626667.2022.2094709

Thanks for the references. We were referring to:

Clark, M. P., Fan, Y., Lawrence, D. M., Adam, J. C., Bolster, D., Gochis, D. J., Hooper, R. P., Kumar, M., Leung, L. R., & Mackay, D. S. (2015). Improving the representation of hydrologic processes in Earth System Models. Water Resources Research, 51, 5929–5956. https://doi.org/10.1002/2015WR017096

Table 1 legend, what is K ?

K refers to the case where soil hydraulic parameters from Kupers et al. 2019 were derived. It's now defined in the Table caption as "K in the experiment name of Case 5 indicates soil property derived from Kupers et al. [2019b] is used."

Most of the end part of section 3.4 needs to use table and figures to support the all statements in the text (text is not using panels from figure 8 which I imagine needs to be included) Here two examples

Thanks! We added table and figures to support the statements in the text of section 3.4.

L582 -587 results presented here need to cite figures or tables where the information contained in the text is shown

Thanks! We cited Fig. 8 in this section.

L594, indicate in the text where is it shown that inclusion of VWC can explain more than 80% of the variance

Thanks! We cited Table 2 in this sentence.

L605 -615 -needs figures or tables to support text

As the model is not good, only performance scores were reported in lines 607 and 608 of the original submission. Figure is not shown as indicated in line 609 of the original submission.

I could not find (figure and table) on which the model is trying to explain the observations.

We clarified in sections where there is only model comparison. Please see our response above in the general comments.

Section 3.4 unclear where the sensitivity is trying to explain spatial variation of modelled AGB or observed AGB.

We now cited tables and figures where appropriate.

The paper has many abbreviations some of which are not defined. Please carefully check they are all explained (including those in tables, i.e table 2) or include a table with all abbreviations.

Thanks! We checked to make sure all abbreviations are defined. Those in Table 2 are added in the caption of Table 2 as shown in the response below.

Table 2: the authors might want to add extra explanation to the reader on how to interpret this busy table.

The caption of Table 2 now reads:

Table 2. Random Forest Model Performance on the simulated above ground biomass (AGB) and water table depth (WTD) from the site wide and 50-ha locations, respectively. Model performance is quantified by mean absolute percentage error (MAPE (%)) and percent of variance explained (VAR$_{ex}$ (%)). The paired data separated by "/" in each column are metrics for training data (left) and unseen test data (right). Subscript RF1 indicates models using topographic features while subscript RF2 indicates model using simulated soil moisture as predictor in addition to the predictors used in RF1 models.

---

## Author Response (AR2)

Dear Editor and Referees,

Thanks again for your valuable comments and suggestions. Attached please find our detailed point-by-point responses (blue texts) to all referees' comments, and changes (purple texts) we made in the revision. We look forward to hearing from you.

Sincerely,

Yilin Fang and Co-authors

**Response to Anonymous Referee #2**

We thank the referee for the suggestions to further improve our manuscript.

Section 3.1 should have a more scientific name (and does not say much), could be the title of the caption of Figure 3. Then it is clear that this section is all a sensitivity analysis. In some instances the text does not refer to simulated AGB or simulated GPP, (lines 479-482) when it should. Best to make sure either refer to simulated or observed fields, so it is clear.

Thanks for the suggestion. The title of Section 3.1 is now changed to "Model sensitivity to lateral flow representation". We clarified in lines 479-482 by referring to "simulated AGB".

Discussions section lines 714-717.. this is all model word..not obs, authors should revise the text and refer to simulations when it should be.

Thanks! We removed lines 714-717 in response to another reviewer's comment.

-Discussion on observed AGB variability : need to include the fact that wood density also affects mortality.

Thanks! We added the following discussion of wood density effect on mortality and the observed AGB variability:

Also not accounted for by the model is the negative relationship between the wood density and tree mortality rate at BCI found in McDowell et al. [2018] using data from Wright et al. [2010]. Including spatially variable tree mortality based on the negative relationship between mortality and wood density may substantially improve model representation of vegetation carbon as indicated by the modeling study in Hancock et al. [2022].

References:

Hancock, M., Sitch, S., Fischer, F. J., Chave, J., O'Sullivan, M., Fawcett, D., and Mercado, L. M. (2022), Modelling the impact of wood density dependent tree mortality on the spatial distribution of Amazonian vegetation carbon, Biogeosciences Discuss. [preprint], https://doi.org/10.5194/bg-2022-87, in review.

McDowell NG, Allen CD, Anderson-Teixeira K, Brando P, Brienen R, Chambers J, Christoffersen B, Davies S, Doughty C, Duque A et al. (2018), Drivers and mechanisms of tree mortality in moist tropical forests, New Phytol, 219(3), 851-869, https://doi.org/10.1111/nph.15027.

Wright SJ, Kitajima K, Kraft NJ, Reich PB, Wright IJ, Bunker DE, Condit R, Dalling JW, Davies SJ, Díaz S et al. (2010), Functional traits and the growth–mortality trade-off in tropical trees. Ecology 91: 3664–3674,  https://doi.org/10.1890/09-2335.1.

Review of 'Modeling the topographic influence on aboveground biomass using a coupled model of hillslope hydrology and ecosystem dynamics'

We thank for the reviewer for the constructive comments and suggestions. Please see our point-by-point responses below.

This study presents a unique effort to link an earth systems model, a hillslope hydrology model and a structured forest and vegetation function simulation model at a well known tropical forest study site in Panama.

Primary findings in my view are that while the model has not been tuned for the site and has unrealistic mortality and biomass response (quantitatively), qualitatively it shows an interesting and promising ability to produce biomass, soil moisture, WTD and biomass turnover gradients over hillslopes. This is the result of coupled hydrological ecosystem and forest regeneration dynamics played out over periods of water stress expected in this seasonal tropical forest, unique outcomes for a unique modeling effort. This in and of itself represents key progress for the field. The value of this work at BCI will only increase from here if further steps are taken to improve model structure, tune the model and compare it more rigorously against the rich datasets available, particularly for biometrics/forest dynamics inside and (if possible) outside of the 50 ha plot. A rigorous validation analysis through time could even be used to test mortality models against one another in a model testing framework (rather than to show, as this study does as a first step, that different water stress mortality formulations have different outcomes).

Thanks for the nice summary of our study and your kind recognition of the promising ability of our modeling of coupled key processes and recommendations for future studies. We added discussions in response to your concerns and suggestions with respect to the boundary conditions, the relevance of the regression analysis, the importance of soil texture and PFT compositions, and places identified for clarifications. Please see the details provided below for each point.

The study may however be limited by a few important pieces, even when recognizing its place as a stepping stone. One is that the primary variation of interest appears to occur between the slope outside of the plot and the state inside the plot, which is relatively homogeneous (an issue familiar to 50 ha plot network research, an advantage of the data for cross site work and perhaps disadvantage for within site work). Nevertheless, this study provides model evidence that including hillslope hydrology creates WTD and associated seasonal soil moisture gradients that can make slopes and topographically lower areas more resilient to water stress, and enhance biomass in these areas. That is interesting and addresses current hypotheses and debate (see suggested consideration of Costa et al 2022 New Phytologist and Sousa et al 2022 GEB references below). Nevertheless, it seemed that this finding--of biomass differences over hillslopes--could have been a greater point of focus for the discussion in the manuscript particularly given that the Mascaro et al 2011 citation appears, which supports this with remote sensing analysis (this citation concludes that there is lidar based evidence with a map of biomass increasing on the slopes). I was also perplexed by the following: Is the spatial correlation between modeled and observed biomass non significant

inside the 50 ha plot for all of the simulations? That might be expected due to low driver variation. However, I would clarify this as a result, and will return to the issue below.

Thank you for the references. We have included them in the discussion and details are provided later in each relevant point below.

The spatial correlation between modeled and observed biomass is not significant inside the 50 ha plot for all of the simulations reported. We made the clarifications and explanation in the revision in Section 3.1:

The spatial correlation between modeled and observed biomass is not significant inside the 50 ha plot for all of the simulations because of homogeneity of the meteorological forcing, soil properties, and gentle topography.

A concern that I saw in relation to the particular hillslope modeled results was the issue of the impermeable boundary conditions/no lateral flow at the boundary. If I understood this, there is no outflow on the edges of the modeled scene, which all fall just below the plateau … ? Is there any chance that the modeled water is unrealistically 'pooling' in the soil, increasing WTD artificially here..? If it is easy to refute this, great! I would add a small statement to such an effect. I know that computation is a limitation (as is geography of the island), but I wondered if a larger scene would serve the analysis better in the future for this reason.

There is no outflow at the outer boundaries of the model domain. We agree that the water table may be artificially increased near the outer boundary because of the imposed boundary conditions. We did an independent test using the water level in Gatún Lake at the outer boundaries and we found that only the results in the grids within 100 m of the outer boundaries are affected. There are locations with shallow water table away from the outer boundaries (Figure 3) that falls in the range with negative AGB relationship with WTD, so our conclusion holds. Future studies will consider a larger domain as suggested. The following statement is added in Section 3.3:

Note that the water table may be artificially increased near the outer boundary due to the no flow assumption, but a sensitivity experiment using the water level in Gatún Lake at the outer boundaries shows no impact on the conclusion that lower areas are more resilient to water stress and has more biomass.

I see the value of the regression tree analysis to summarize and capture the predictability of drivers of the simulation outcomes. However, I felt that the specific details and discussion could have been de-emphasized relative to other issues, such as those related to improving the model realism to improve predictions across demographic, flux, and soil moisture variables. Specific RT results are not likely to remain of interest if the model is improved, suggesting that the focus should be on what RT analysis can reveal and how eventually it can complement empirically driven analysis.

Thanks for the comment. The purpose of the regression tree analysis is not to summarize and capture the predictability of drivers of the simulation outcomes. We used the analysis to determine if similar relationships between the selected features (topography, soil water state, etc.) and AGB can be established from the simulations and from the observations in the 50-ha plot. For example, if

the RF model indicates that the observed AGB can be explained by the soil water state but the simulations do not capture such relationship, we can then focus on how to improve simulation of soil water state and its relation with AGB using the coupled model. On the other hand, if neither the observed nor simulated AGB can be explained by the selected features, our model needs to account for processes that are not currently represented (e.g., nutrient gradients). We made changes in the Introduction as shown below to further clarify the purpose of the RT analysis and in the Discussions and Conclusions section, we removed the discussion of RF that was built based on the modeling results and focused only on what we learn from the RT analysis of observed AGB with selected features, which highlights that factors other than the selected features may play more important roles in controlling AGB in the study site.

The purpose of the RF models is to reveal whether there are similar nonlinear relationships between topography, soil water states, and AGB in the coupled simulations and in the observations. This analysis may reveal model limitations in capturing certain nonlinear relationships found in the observations and inform future efforts to improve modeling of coupled hydrology-vegetation processes.

The discussion of the dependency on water stress mortality kernels ended up somewhat superficial relative to my expectation from the presentation of the different models (which included some specification and description of the mechanisms). Why was it the M1 mortality function that led to such relatively weak dependency of AGB on water table depth while the others showed such drastic (and likely unrealistic) dependencies? That these models have different impacts overall is not surprising.

The weak dependency of AGB on water table depth with the M1 mortality function is because the plant wilting factor (Eq. 12) calculated at the site is much larger than the prescribed threshold of $10^{-6}$, which results in no hydraulic failure induced mortality. The drastic dependencies of AGB on water table depth for the M2 and M3 mortality function seems to be unrealistic, other functions or thresholds to trigger the hydraulic failure should be explored.

The following sentence is added to explain the effect of the M1 mortality function and point out the drastic AGB dependencies with M2 and M3:

AGB from ELM-PF-F1-M1 is the least sensitive to water table depth because the plant wilting factor (Eq. 12) calculated at the site is much larger than the prescribed threshold of $10^{-6}$, which results in no mortality due to hydraulic failure.

However, the fast decrease of AGB with WTD using the M2 and M3 functions seems to be unrealistic and requires future exploration.

What is the role of vertical soil water stratification and how was this implemented in the model? In a few cases it seemed averages over vertical soil water profiles were taken for analysis (but not totally clear). I realize that this is not a focus and thus does not justify great expansion, however, the paucity of discussion seemed incomplete with negative consequences for understanding. For example, FATES hydraulic redistribution capacity seemed potentially important, which made sense

to me when mentioned I think only because I know a little bit about FATES structure. I would suggest a clear sentence or two about vertical soil gradients in the methods. Related to this, it was unclear how the model produced water table depth predictions and how this was related to soil moisture. Finally, I will add that while it seems not a direction of this study at the moment, a lot of attention is being given to vertical stratification of water uptake, which may differ by tree size and function strategy (e.g. Chitra-Tarak et al 2021. New Phytologist)

The vertical soil water stratification is solved by the model from the mass balance equation in Eq. 1 and the mass equation of water from the soil to the plants and atmosphere in FATES-hydro. When discretized numerically into grids in 3D space, Eq. 1 equates the time rate of change of water mass within a grid with the mass fluxes of water across the surfaces of each grid and water source/sink. This results in a matrix equation including every grid, both horizontally and vertically. The water table is the surface where the water pressure head is equal to the atmospheric pressure. It was calculated by the hydraulic head of the first water saturated (i.e., the soil moisture equals the porosity) soil layer from the ground surface. Similarly, a matrix equation is solved for FATES-hydro, which automatically takes into account of the vertical stratification of water uptake depending on the total potential between the soil and the plant.

We added the following statement for clarification in the revision in Section 2.1.2:

When discretized numerically into grids in three dimensions, Eq. 1 equates the time rate of change of water mass within a grid with the mass fluxes of water across the surfaces of each grid as well as water source/sink. This results in a matrix equation including every grid, both horizontally and vertically. The water table is the surface where the water pressure head is equal to the atmospheric pressure. The surface was calculated by the hydraulic head of the water saturated (i.e., the soil moisture equals the porosity) grid near the ground surface.

The vertical grid resolution was added in Section 2.4:

The grid resolution for ParFlow in the x and y directions is 90 m and varies from 7 mm (near the ground surface) to 35 m (near the bedrock) in the z direction.

Soil structure is a huge issue that was not clearly addressed in the 'factors limiting inference of field data' discussion. Soil structure is an essential complexity related to hillslopes in tropical forests. In much of the tropics uplands are more clayrich than lowlands, which are sandier or siltier in general. I do not know about BCI. This has a big effect on flow dynamics obviously. See discussion of this in the Costa et al 2022 paper.

Thanks! We added the following sentence in Section 4 of the revision:

It is necessary to have a better quantification of the soil texture and related hydraulic properties as the distribution of biomass is the combined result of plant traits, soil properties, climatic and groundwater conditions [Costa et al., 2022]. AGB can be influenced by soil texture which directly affects the time interval between precipitation inputs and groundwater recharge [Sousa et al. 2022] and the capillary fringe above the water table that supplies water to the rooting zone

[Costa et al. 2022]. For example, results in Sousa et al. [2022] suggest a contribution of clayey texture in increasing AGB in dry climates with a shallow water table.

In general, I appreciated that the discussion was concise, but I would have foregone some discussion of the Random Forest results and focused on what is needed next to make the model produce more realistic results! The RF model does make clear and interesting points, but would have been stronger if it was comparable to, or could be extended to, the empirical data. Perhaps utilizing the broader lidar derived estimates of Mascaro et al 2011 (or another updated effort) would offer some hope for this approach in the future. As it stands, the detailed discussion of RF would be more satisfying if the model were performing more realistically. I am not sure what to take away from the relative importance comparisons without such grounding.

We agree with the reviewer that the RF analysis would have been more useful if we were able to develop a generalizable RF model that performs well using both training data and test data based on observations. This way we can evaluate whether our simulations are able to capture the relationships revealed from observations, which is the main goal of the RF analysis. Please see our previous response clarifying the purpose of the RF analysis. As we are not able to develop a generalizable RF model based on observations, as likely limited by the amount of observation data, we have significantly shortened the discussion of the RF analysis in the Discussions and Conclusions section (Section 4) in the revision.

The plant functional type comparison is a nice additional component. It is understandable to start with something simple like a few present functional groupings (PFTs). That said, it would perhaps be worth noting that these should, in the future, also include specifications for differences in water stress responses. There are citations suggesting that tropical pioneers are more hydraulically vulnerable (cited/discussed in Costa et al 2022). Generally, this is necessary to get the best quantitative mortality response, and incorporate it into the model to produce ultimately realistic mortality rates. On that point, I think that M2 and M3 just are sending mortality too high too fast with soil water stress, and that is driving your rapidly asymptoting to too low level AGB with increasing WTD responses. Trait/PFT composition will impact these rates and should be a key part of advancing these models too.

Thanks for the suggestion and reference. We included your comments in the revised discussion as the following:

As a demonstration, only two plant functional types were considered in this study. When water stress is considered, the negative response of AGB with WTD simulated by the model is supported by previous studies (e.g., Esteban et al., 2021) that species associated with deep water tables had decreased growth and increased mortality compared to those associated with shallow water table depth during severe drought. However, the two hypothetical hydraulic failure models (M2 and M3) result in a strong positive relationship between mortality with the soil water stress, driving unrealistic response of AGB to increasing WTD. In reality, the resistance of trees to water stress also depends on the severity of droughts, plant traits, and environmental conditions [Costa et al., 2022; and references therein]. For example, previous studies found that

hydraulically-vulnerable trees can delay dehydration by accessing deep water during droughts in BCI [Chitra-Tarak et al., 2021]. How plant traits and PFT composition will impact these rates should be a key consideration in advancing coupled modeling in the future.

Additional reference:

Esteban, E. J. L., C. V. Castilho, K. L. Melgaco, and F. R. C. Costa (2021), The other side of droughts: wet extremes and topography as buffers of negative drought effects in an Amazonian forest, New Phytol, 229(4), 1995-2006, doi:10.1111/nph.17005.

A few more specific points…: There were some shortcuts in model description that would only be clear to insiders; e.g., talking about target biomass, carbon storage, etc. That would be out of the blue and not make much sense to someone that had no knowledge of flavors of ED I think… I would try to include some descriptive topic sentences about how that is important for determining plant relative performance and is impacted by resources and competition, etc. when first mentioned

Thanks for the comment. We included descriptive sentences before "target biomass and carbon storage" were first mentioned as below to put them in a meaningful context:

At the daily time step, daily carbon increment calculated in FATES is sequentially allocated per cohort [Koven et al., 2020]. The amount is subtracted from the cohort's storage pool if the carbon increment is negative. If the carbon increment is positive, the cohort first replenishes the carbon storage pool and tissue turnover is then compensated. The cohort will allocate the remaining increment to any organ pools (leaf, stem, coarse root, fine root, and seed) that are below their allometric targets. The cohort will grow its stem diameter, allocating to each pool proportionally to that pool's derivative with respect to stem diameter using the remaining carbon increment (if any).

Why would different mortality functions not impact AGB variation as is suggested?

The M1 mortality function does not impact AGB variation because the stress factor is larger than the prescribed threshold (see our response to an earlier comment). M2 and M3 affect the AGB variability in the whole domain, but not much in the 50-ha plot because of the relatively homogeneous soil hydrology there. This statement is added in Section 3.3 in the revision.

I have not said much about the time series of ecosystem function components. That is a nice analysis, and a benefit for the paper. A note though, I would be a little careful in explaining what this is and how it operates clearly in your model experiment description. It took me a little while to understand the need for an extra 16 years of run time (i.e. when the met. drivers were available, right?). Maybe I missed something earlier but this section required a little head scratching before I realized that this pertained to comparison of time series ecosystem data.

We apologize for the lack of clarity. The 16 years of simulation was chosen such that the meteorological forcing aligns with the years of observation. This sentence was added in the revision for clarification.

In sum, I have raised some concerns and suggested action points for improvement. I want to step back though and say that this manuscript encompasses a nice progress report on new modeling developments and applications in the field. I just think that shifting the focus a little could do a better job to highlight what this paper is showing effectively that is novel and good, and should be further developed, i.e., that we can model/capture hillslope function and biomass responses. This will, in my view, entail stepping away from some detail about the specific predictions of RF that will end up less relevant when a better version of the model is built and running. I really hope that such future work is a serious plan; this could bring great leaps towards model-based testing for model structure comparisons. Furthermore, in discussion, I think there could be fewer generalities about the merits of such modeling, and more about how this will be improved by addressing its limitations for BCI or other tropical sites (while addressing workers rights).

Thank you very much for your kind words, concerns and suggestions, which help further improve our manuscript. We removed the discussion of RF for the models in response to this comment and an earlier comment above. We have work in progress using our coupled model at another site (Manaus), where there are ongoing water and vegetation related measurements along the transect of a hillslope for us to evaluate the hillslope function and biomass responses.

Some more points on potentially overlooked citations:

Thanks for the references. We added them in the revision.

Sousa, T.R., Schietti, J., Ribeiro, I.O., Emílio, T., Fernández, R.H., ter Steege, H., Castilho, C.V., Esquivel-Muelbert, A., Baker, T., Pontes-Lopes, A. and Silva, C.V., 2022. Water table depth modulates productivity and biomass across Amazonian forests. Global Ecology and Biogeography.

Costa, F.R., Schietti, J., Stark, S.C. and Smith, M.N., 2022. The other side of tropical forest drought: do shallow water table regions of Amazonia act as large-scale hydrological refugia from drought?. New Phytologist.

Mascaro et al 2011 is cited but the finding that AGB increases on BCI hillslopes is surprisingly not mentioned as far as I noted. Instead it is stated that biomass spatial structure is not known for evaluation in the results section. I am a bit confused by this. Within the 50ha plot it certainly is known, and this lidar analysis suggests that there is a ready source for exploration in the literature for quantitative BCI hillslope biomass effects.

The variability explained by the multiple regression approach in Mascaro et al [2011] is 14% and 33% depending on the resolution, which is about the same skill as the RF model for our training data. We added the following sentence to include the finding from this reference:

Even though it finds that slope is an important driving factor in the training data of the observed AGB, as supported by Mascaro et al. [2011] using the multiple regression method to examine controls over AGB derived from airborne Light Detection and Ranging (LiDAR) at BCI, it cannot master the test data (negative explained variance).

Yes, the spatial biomass structure at BCI is known, which has been indicated in our results and discussions. The first sentence in section 3.3 "As there is no spatial observation of the relationship between AGB and WTD at the site" refers to no spatial obversion of WTD. We changed the sentence to "As there is no spatial observation of WTD at the site" for clarification.

Chitra-Tarak, R., Xu, C., Aguilar, S., Anderson-Teixeira, K.J., Chambers, J., Detto, M., Faybishenko, B., Fisher, R.A., Knox, R.G., Koven, C.D. and Kueppers, L.M., 2021. Hydraulically-vulnerable trees survive on deep-water access during droughts in a tropical forest. New Phytologist, 231(5), pp.1798-1813.